

# Understanding and simulating cropland and non-cropland burning in Europe using the BASE (Burnt Area Simulator for Europe) model

Matthew Forrest[1,], Jessica Hetzer[1], Maik Billing[2], Simon P.K. Bowring[3,4], Eric Kosczor[5], Luke Oberhagemann[2,6], Oliver Perkins[7,8], Dan Warren[9,10], Fátima Arrogante-Funes[11], Kirsten Thonicke[2], Thomas Hickler[1,12]

[1] Senckenberg Biodiversity and Climate Research Centre (SBiK-F), Frankfurt am Main, Germany
[2] Potsdam Institute for Climate Impact Research, Member of the Leibniz Association, Potsdam, Germany
[3] Laboratoire de Géologie, Département de Géosciences, Ecole Normale Supérieure (ENS), Paris, France
[4] Laboratoire des Sciences du Climat et de l'Environnement (LSCE), IPSL-CEA-CNRS-UVSQ, Université Paris-Saclay, Gif-sur-Yvette, France.
[5] Institute of Photogrammetry and Remote Sensing, Technische Universität Dresden, Dresden, Germany
[6] University of Potsdam, Potsdam, Germany
[7] Department of Life Sciences, Imperial College London, The Leverhulme Centre for Wildfires, Environment and Society, London, UK
[8] Department of Geography, King's College London, London, UK
[9] Gulbali Institute, School of Agricultural, Environmentall, and Veterinary Sciences, Charles Sturt University, Thurgoona, Australia
[10] Environmental Science and Informatics Section, Okinawa Institute of Science and Technology, Onna-son, Okinawa, Japan
[11] Department of Geology, Geography and Environment, Universidad de Alcala, Alcalá de Henares, Spain
[12] Department of Physical Geography, Johann Wolfgang Goethe University of Frankfurt, Frankfurt, Germany

*Correspondence to*: Matthew Forrest (matthew.forrest@senckenberg.de)

**Abstract.** Fire interacts with many parts of the Earth system. However, its drivers are myriad and complex, interacting differently in different regions depending on prevailing climate regimes, vegetation types, socioeconomic development, and land use and management. Europe is facing strong increases in projected meteorological fire danger as a consequence of climate change, and has experienced extreme fire seasons and events in recent years. Here, we focus on understanding and simulating burnt area across a European study domain using remote sensing data and Generalised Linear Models (GLMs). We first examined fire occurrence across land cover types and found that all non-cropland vegetation types (NCV, comprising 26% of burnt area) burned with similar spatial and temporal patterns, which were very distinct from those in croplands (74% of burned area). We then used GLMs to predict cropland and NCV burnt area at ~9x9 km and monthly spatial and temporal resolution, respectively, which together we termed BASE (Burnt Area Simulator for Europe). Compared to satellite burned area products, BASE effectively captured the general spatial and temporal patterns of burning, explaining 32% (NCV) and 36% (cropland) of the deviance, and gave similar performance of state-of-the-art global fire models. The most important drivers were fire weather and monthly indices derived from gross primary productivity, followed by coarse socioeconomic indicators and vegetation properties. Crucially, we found that the drivers of cropland and NCV burning were very different,



highlighting the importance of simulating burning in different land cover types separately. Through the choice of predictor variables, BASE was designed for coupling with dynamic vegetation and Earth System models, and thus enabling future projections. In particular, the strong model skill of BASE when reproducing seasonal and interannual dynamics of NCV burning (i.e. temporally evolving wildfire risk), and the novel inclusion of cropland burning, recommend it for this purpose. In addition to this, the BASE framework may serve as a basis for further studies using additional predictors to further elucidate drivers of fire in Europe. Through these applications, we suggest BASE may be a useful tool for understanding, and therefore adapting to, the increasing fire risk in Europe

## 1 Introduction

Fire is recognised as a fundamental ecological force (McLauchlan et al., 2020), a key component of the Earth system (Archibald et al., 2018; Bowman et al., 2009), and a serious hazard for human health, livelihoods, property, wildlife and biodiversity (Arrogante-Funes et al., 2024; Bowman et al., 2020; Johnston et al., 2012; Sullivan et al., 2022). It interacts with many components of the Earth, affecting biogeochemical cycles, alters surface energy budgets, influences vegetation dynamics and composition as well as soil ecology, and alters the chemical composition of the atmosphere, thereby influencing regional and global climate (Archibald et al., 2018; Bowman et al., 2009; Jones et al., 2022) and human health. Total global burnt area (including fires set deliberately for land management) is decreasing, primarily driven by decreases in fire in savanna, grassland and cropland regions (Andela et al., 2017). However, the frequency of extreme wildfires is increasing (Cunningham et al., 2024), as is forest area loss due to fire (Tyukavina et al., 2022). Many regions are experiencing wildfires of hitherto unrecorded extent and/or severity, e.g., 2019/2020 in Australia (Boer et al., 2020) and 2023 in Canada (Hu et al., 2024). Studies of regional fire dynamics can help resolve these complexities by revealing region-specific processes and drivers, whilst also providing results which can inform policy at a coherent political level over a broad spatial extent. One such region is Europe, which is experiencing unprecedented wildfires (San-Miguel-Ayanz et al., 2023). The already fire-prone region of southern Europe has been experiencing extreme fire seasons with difficult to control fires, for example in Portugal 2017 (Turco et al., 2019), Greece 2018 (Giannaros et al., 2022), and southwestern Europe 2022 (Rodrigues et al., 2023). In northern and central Europe, regions which were not previously considered fire-prone are now experiencing wildfires (Arnell et al., 2021; Krüger et al., 2023). Even moderate climate change scenarios show large increases in fire danger due to fire weather changes (El Garroussi et al., 2024; Turco et al., 2018). Thus there is an urgent need to understand and simulate fire occurrence at the European scale.

However, whilst the basic physical prerequisites of fire occurrence can be summarised fairly simply as: a sufficient amount of spatially-continuous, suitably-aerated, dry fuel and an ignition source, understanding where and when these conditions are fulfilled and how large the resulting fires becomes is rather more complex. Meteorological conditions ("fire weather") at the



time of a fire affects its rate of spread and conditions in antecedent days affect the moisture content of both live and dead fuels.
Vegetation, the primary fuel source, varies tremendously across the planet resulting in large heterogeneity in fuel conditions both in terms of fuel moisture and physical flammability characteristics (ie. leafy vs woody, dead vs live fuel, fuel particle dimensions). Human activity and infrastructure account for the majority of fire ignitions (responsible for 96% of burnt area in Europe, Dijkstra et al., 2022) but lightning and other natural ignitions also occur. Humans may start fires for myriad reasons (including negligence and arson) which will vary depending on land use type and cultural practices, but humans also work to
suppress fires (Millington et al., 2022). Legislation and law enforcement also play a role if fire practices are allowed to manage the landscape or how well fire-fighting techniques are funded and can be applied. Land use also affects the vegetation, and hence fuel conditions, and introduces barriers to fire spread into the landscape. Topography affects rate of spread and can also introduce barriers to fire spread. In summary, we find a plethora of factors affecting fire occurrence, and expect them to function differently depending on the local vegetation, socioeconomic development and human activity.


Two modelling approaches have typically been used to study fire occurrence from an Earth system perspective or at large scales. Process-based fire models coupled to land surface and Dynamic Global Vegetation Models (DGVMs), have been used for studying fire dynamics by simulating biophysical mechanisms and some socioeconomic factors across a range of complexities (Hantson et al., 2016). Complementary approaches using correlative methods have been developed using either
statistical models (such as Generalised Linear Models, GLMs, for example Bistinas et al., 2014; Haas et al., 2022) or machine learning models (typically random forests, Forkel et al., 2017; Kuhn-Régnier et al., 2021; Mukunga et al., 2023). These approaches typically use a larger set of input variables, include more socioeconomic variables, and use observed vegetation. Both types of models are usually applied at global scope and so are inherently focussed on matching the global pattern of burnt area. This global pattern is dominated by grass fires in the tropics, particularly Africa, and the fire-enabled DGVMs do a
reasonable job simulating this (Hantson et al., 2020). However, they have notable regional discrepancies, likely because their global focus leaves them unable to resolve regionally-specific processes or phenomena. On the other hand, national and sub-national scale studies are inherently limited in the range of environmental and socioeconomic conditions that they encompass, and hence in their broader applicability. Thus there is a need to develop models focussed on intermediate, (i.e. continental) scales (Boulanger et al., 2018; Keeping et al., 2024; Turner et al., 2011) where there is large variation (and hence applicability),
but the patterns are not overpowered by the tropical savannas and the model is sensitive to regionally important phenomena.

Concerning fire regimes at the pan-European scale, recent studies have described and quantified fire regimes in Europe (Galizia et al., 2021); estimated the fractions of anthropogenic vs lightning ignitions and their contribution to burnt area (Dijkstra et al., 2022); investigated drivers of large to extreme fire occurrence in terms of individual events (Ochoa et al., 2024); and examined
the compounding effects of fire and other hazards (Sutanto et al., 2020). Other research has focussed on relating burnt area to fire weather variables in different regions of Southern Europe e.g. districts of Portugal (Carvalho et al., 2008), NUTS3 subregions (Turco et al., 2018), and Mediterranean countries (Amatulli et al., 2013). There have also been a number of DGVM





studies which used global fire models to project future changes in burnt area in Europe but which do not focus specifically on the driving factors and have only limited regional adaptation for Europe (Dury et al., 2011; Migliavacca et al., 2013; Wu et al.,

2015; Khabarov et al., 2016). We are not aware of any study examining drivers of burnt area which simultaneously (i) considers specifically the pan-European scale, (ii) considers drivers beyond fire weather related variables, and (iii) operates at a gridded resolution for integration with DGVMs and which is necessary for including highly heterogeneous topographic, socioeconomic and vegetative factors. Furthermore, Europe has a diverse array of land cover types and these have not been distinguished in previous studies. This is of particularly importance given recent advances in satellite observation of burnt

area which indicate higher than previously estimated occurrences of fire in croplands (Hall et al., 2024; Roteta et al., 2019)

Here we seek to fill this knowledge gap by disentangling the drivers of fire occurrence across a European study domain (defined here as the European Union 27 plus the United Kingdom and 6 Balkan candidate countries). This study's aims are twofold: (i) to gain insight into drivers of fire activity across land cover types in Europe, and (ii) to encapsulate this knowledge into a new

fire model that can be embedded in a DGVM. To fulfil these aims we chose to use GLMs. As a correlative method, GLMs have the advantage over process-based models that they are highly data driven and so can tease out process understanding rather than only embodying existing knowledge. But also, compared to more complex correlative techniques (for example random forests), GLMs can be described by a handful of coefficients and so can easily be embedded within other models. As a preamble to developing the GLMs we first examined fire and landcover data to determine which broad land cover categories

should be simulated. We then fitted GLMs to determine which environmental and socioeconomic variables can explain fire behaviour in Europe and produce parsimonious predictive models.

## 2 Materials and Methods

### 2.1 Datasets

This study relied solely on gridded datasets, with the common spatial resolution determined by a state-of-art climate dataset

with 0.07(03135)°, which corresponds to approximately 9 x 9 km, derived from ERA5-Land (Muñoz-Sabater et al., 2021). This dataset was selected in order to provide comparatively fine spatial resolution and compatibility with the FirEUrisk Assessment System (Chuvieco et al., 2023), and is available until 2014. For clarity, we refer to elements of the target 9 km grid as *grid cells*, and elements of higher resolution grids as *pixels*. Unless otherwise noted, all data processing was done using R (R Core Team, 2024) and the terra package (Hijmans, 2023).

### 130  2.1.1 Fire occurrence and land cover combination

Central to this analysis was the combination of ESA FireCCI51 (Lizundia-Loiola et al., 2020) and ESA LandcoverCCI (ESA, 2017) datasets, which we used to quantify fire occurrence in different land cover types (LCT) in two different ways: burnt area, *BA* (ha), and burnt fraction, *BF* (unitless) on a monthly basis. We also calculated the fraction of a gridcell covered by an





LCT, *LF* (unitless). To combine these products we first performed nearest-neighbour regridding to bring the 300m land cover

data on to the 250m grid of the burnt area data. Then, for a given LCT, month and 9 km gridcell, we calculated $BP_{LCT}$, the

number of burnt pixels in the gridcell (from the FireCCI51 pixel product land cover layer), and $TP_{LCT}$, the total number of

pixels of that LCT (from the regridded LandcoverCCI product). We also calculated the total area of the 9 km gridcell, *A* (ha),

and the number of 250m pixels within that 9 km gridcell, *TP*.

We calculated burnt fraction (unitless) as:

$$BF = \frac{BP_{LCT}}{TP_{LCT}} \tag{1}$$

Land cover type fraction (unitless) was calculated as:


$$LF = \frac{TP_{LCT}}{TP} \tag{2}$$

And finally burnt area (ha) as:

$$BA = BF.LF.A \tag{3}$$

Note that this method accounts for the variation in gridcell size with latitude, but not the (far smaller) variation of pixel size

within a 9 km gridcell.

Burnt fraction was used as the target variable for model fitting and mean burnt fraction (averaged across grid cells) was used

for comparing temporal patterns of fire occurrence between LCT. Burnt area was used for comparing predicted fire occurrence

to the observed patterns, both in terms of agreement metrics and visualisation, and comparing the overall amounts of burnt

area present in the study area.

### 2.1.2 Fire weather and wind speed

To capture fire weather we used an adapted version of the Canadian Forest Fire Weather Index (FWI) (Van Wagner, 1987)

that considers the total daily precipitation combined with the daily temperature, relative humidity, and wind speed at noon.

Here we calculated it using the implementation of the Canadian Forest Fire Danger Rating System in the R package "cffdrs"

which calculates the FWI and all subindices (Wang et al., 2017b). The climate variables required were taken from a version

of the ERA5-Land climate datasets which was produced by regridding the original triangular–cubic–octahedral (TCo1279)

operational grid from the reanalysis simulations (Muñoz-Sabater et al., 2021) to a regular 9 km grid (~0.07°) across Europe



(Chuvieco et al., 2023) in order to maintain a higher spatial resolution than the standard 0.1° resolution. We used accumulated daily precipitation (in mm), the noon values were approximated using the maximum daily temperature (in C°), the minimum relative humidity (in %) and the daily mean wind speed (in km/h) by the approach of Hetzer et al (in prep.). Monthly averages were calculated from the daily FWI values. We also considered the monthly mean and maximum of wind speed from this climate dataset as candidate predictors.

### 2.1.3 Gross primary productivity and derived quantities

We considered gross primary productivity (GPP), and quantities derived from it, as potential predictors for fuel accumulation and ecosystem state. The monthly version of GOSIF GPP product (Li and Xiao, 2019) was regridded from its native 0.05° resolution to the target grid using average resampling. From these monthly values we calculated the sum of the antecedent 12 months (GPP12) following (Kuhn-Régnier et al., 2021) to quantify fuel build up. We also derived two indices to quantify ecosystem state and post-harvest timing (only for use in the cropland burning model). We define the monthly ecosystem productivity index (MEPI) as this month's $GPP_m$ divided by the maximum of the 13 previous months (including this month), i.e.

$$MEPI = GPP_m \Big/ \max(GPP_m, GPP_{m-1}, \dots, GPP_{m-12}) \tag{4}$$

MEPI therefore ranges between 0 and 1. High values indicate that photosynthesis is occuring at close to its maximum rate and so the ecosystem is in an unstressed state with full leaf expansion - i.e. high proportions of live fuel and high live fuel moisture content and thus low expected flammability. Low values imply either a dormant state (i.e. leaves senesced and higher dead fuel proportion) or a stressed state, which we expect to correspond to higher flammability.

We defined the post-harvest index (PHI) by,

$$PHI = \text{mean}(GPP_{m-1}, GPP_{m-2}, GPP_{m-3}) \Big/ \max(GPP_m, GPP_{m-1}, \dots, GPP_{m-12}) \tag{5}$$

The logic behind PHI is that crop residue burning is likely to happen when productivity for the previous three months has been high, relative to the annual maximum. Such a situation indicates a productive growth period for the crops, after which point the crops can be harvested, creating an opportunity for residue burning. Note that we expect the opposite response for PHI compared to MEPI, with high values of PHI indicating a higher likelihood of fire occurrence but low values of MEPI indicating higher likelihood.



### 2.1.4 Fraction of absorbed photosynthetically active radiation

The fraction of absorbed photosynthetically active radiation (FAPAR) can be used to quantify fuel buildup and availability

(Forkel et al., 2017; Knorr et al., 2016; Kuhn-Régnier et al., 2021). Here we used the FAPAR 1km v2 product by the Copernicus Global Land Service (CGLS), which is derived from SPOT/VEGETATION and PROBA-V data (European Commission Directorate-General Joint Research Centre, 2020). It is originally provided at 1km resolution globally but was aggregated and regridded to the 9km target grid using pixel averaging and bilinear interpolation. It covers the analysis period in 10-day steps until June 2020. For each timestep, the final consolidation product RT6 was used.

### 2.1.5 Tree cover

The degree of tree cover affects fuel load and composition, local wind speed, and fuel moisture (due to subcanopy microclimates). A recent global study indicated that tree cover has a negative effect on burnt area (Haas et al., 2022). For maximum precision we used the global 30m Landsat tree canopy version 4 product (Sexton et al., 2013) which was processed to 9 km resolution by simple averaging. We took the mean of the layers for 2000, 2005, 2010 and 2015 to smooth out

occasional artefacts seen in the individual layers.

### 2.1.6 Population density and gross domestic product

The presence of humans has long been recognised as affecting fire occurrence and population density, and is widely used in global fire models (Hantson et al., 2016; Rabin et al., 2017) and empirical studies of global fire patterns (Bistinas et al., 2014; Haas et al., 2022). For this study, population density and GDP data was acquired from the HYDE v3.2 database 'baseline'

version (Klein Goldewijk et al., 2017) . This was converted from ascii to netCDF format using GDAL (GDAL/OGR contributors, 2023). Annual maps were created by linearly interpolating between the five-yearly population density estimates, performed using Climate Data Operators (Schulzweida, 2023), and remapped to this study's ~9 km spatial resolution.

### 2.1.7 Human development index and gross domestic product

Both human development index (HDI) and gross domestic product (GDP) have been used as socioeconomic indicators to

represent human effects on fire regimes. (Li et al., 2013) implemented a suppression of both non-cropland and cropland fires with increasing GDP per capita in a global fire model. More recently, (Chuvieco et al., 2021) used HDI in an analysis to explain variability in burnt area and found that increasing HDI dampens burnt area interannual variability. Here both HDI and GDP per capita were taken from the datasets from Kummu et al (Kummu et al., 2018) and regridded to the target 9 km resolution by simple averaging. This dataset introduced the small data gap in northern Macedonia.






### 2.1.8 Topographic variables

The interactions between terrain and fire spread can be highly complex and variable (Sharples et. al, 2009). On the one hand, rough terrain can be expected to increase fire size by increasing fire spread rate at a local scale on slopes (e.g., Rothermel et al., 1972) and by limiting access to fire fighters. On the other hand, it may reduce fire size by introducing barriers to fire spread. In their study at 0.5° resolution, (Haas et al., 2022) found that the vector ruggedness measure, a measure of terrain roughness, had a negative effect on burned area in a grid cell. In contrast, the topographic position index (TPI), which quantifies the relative proportions of hill tops to valley floors, was found to have a slightly positive effect. Here we extracted a set of terrain variables from the Geomorpho90 dataset, which is based on the 90m resolution MERIT digital elevation model (Amatulli et al., 2020). We masked out pixels with more than 50% urban or permanent resolution based version 3 of the Copernicus Land Cover dataset (European Commission Directorate-General Joint Research Centre, 2020) as such pixels are not expected to burn and influence fire behaviour. We then aggregated these to the target grid by calculating the median of pixels using Google Earth Engine (Gorelick et al., 2017). We found that at our target resolution of ~9 km, all the terrain variables fell into two groups of strongly correlated variables (data not shown). From these groups we picked slope and TPI because of their relative simplicity of interpretation and for comparability with other studies.

### 2.2 Analysis of fire occurrence by land cover types

Before performing the main task of building GLM models, we first grouped land cover types based on their relative contributions to the total burnt area and their spatiotemporal patterns of burning. We therefore examined fire occurrence in land cover types using the ESA LandcoverCCI dataset. We first of all separated cropland from non-cropland areas to form two main land cover types, and then divided these types into subtypes. For cropland we considered the following subtypes: "herbaceous croplands" (combining classes 10, 11, and 20); "woody croplands" (class 12 only) and "mosaic croplands" (comprising the two mosaic cropland-natural types, classes 30 and 4). For non-croplands types we took the "grasslands" subtype (class 130) by itself; combined the three shrubland categories (classes 120, 121, 122) into a single "shrublands" category; combined all the tree-dominated categories (classes 50, 60, 61, 62, 70, 71, 72, 80, 81, 82 and 90) into a single "woodlands" category; and formed a "natural mosaics" category from the two natural mosaics classes (100 and 110). We also analysed "sparse vegetation" (classes 150, 151, 152 and 153) distinctly from the other vegetation types.

We then compared the mean annual burnt area in each of our aggregated classes to determine their relative contributions to fire occurrence in Europe, indicating which classes are most important to simulate. To determine how we might group the subtypes we examined the spatial patterns of burnt area, and the interannual variability and season cycle of mean gridcell burnt fraction. Based on this, we concluded that it would be sufficient to build separate models for only two land cover types: croplands (excluding woody and mosaic cropland types) and non-cropland vegetation (hereafter NCV).





| Quantity | Reason | Temporal resolution | NCV | Cropland |
|---|---|---|---|---|
| Fire Weather Index (FWI) | Fire weather conditions | Monthly | Log, interaction with FWI | Linear |
| MEPI | Monthly Ecosystem Productivity Index - health and phenological state of vegetation | Monthly | Interaction with MEPI | Linear |
| PHI | Post harvest index | Monthly | - | Linear |
| Windspeed | Affects rate of spread | Monthly | - | Quadratic |
| Human Development Index (HDI) | Socio-economic, proxy for cultural practices, investment in firefighting, public awareness and legislation | Annual | Linear | Linear |
| GDP | Socio-economic | Annual | - | - |
| Pop_dens | People start/extinguish fires | Annual | Square root | Square root |
| FAPAR12 | Fraction Absorbed of Photosynthetically Active Radiation - fine fuel build up over last 12 months, general productivity | Past 12 months | Linear | - |
| GPP12 | Gross Primary Productivity- fine fuel build up over last 12 months, general productivity | Past 12 months | - | Quadratic |
| Treecover (%) | Fuel characteristics and ecosystem openness | Static map | Quadratic | - |
| Slope | Topographic: affects rate of spread, fragmentation, access | Static map | Linear | Linear |
| Topographic Position Index (TPI) | Topographic: affects rate of spread, fragmentation, access | Static map | Linear | - |

**Table 1. List of all predictors variables considered for BASE; the reasons for their inclusions; their temporal resolution; and the form of the associated term in the final BASE models (including pre-applied transformations of log and square root)**



## 2.3 GLM fitting

We fitted GLMs for the NCV and cropland LCTs using the standard 'glm' function is R over the period 2002-2014 (determined by the climate dataset). The quasibinomial family was used to account for the high degree of overdispersion (large amount of zero values) in the data with the logit link function. Note that the use of a 'quasi' family precluded the use of some standard GLM tools such a QQ-plots and information criteria because there is no clear generating model. We also chose not to scale the predictors in order to maintain maximum interpretability of the results, but our testing showed that scaling made no difference to model fit results.


We considered every month and gridcell which had more than 10% of the LCT present as a data point, and used 80% of the data points for training and kept 20% for testing. When comparing the normalised mean error (NME, Kelley et al., 2013) between the testing and training we saw differences of ~0.002.

## 2.4 Predictor variable selection

For predictor variable selection we took an approach that could be summarised as "process-informed trial-and-error". This was chosen over automated variable selection methods because we wanted to select and test specific variables to capture specific effects or processes. This means, for example, that at least one variable that is an indicator of fuel availability and one for fire weather, must be maintained in the model. As the model was developed, variables were added, substituted or removed. Interaction terms and different responses (e.g. quadratic terms) were also tested. This required continuous evaluation of model 280 performance and of the responses of individual variables. We also minimised the degree of correlation between predictors by testing only one predictor from any set of highly correlated variables (say FAPAR and GPP, or HDI and GDP) at a time (for correlations of predictors see Figs. B1 and B2). Automated variable selection does not allow this flexibility, nor does it allow informed decision making anchored in process understanding. We therefore present the outcome of this variable selection *fait accompli,* but we also present a table of sensitivity of model results with predictors changed/removed and plots of the effects 285 of removing certain key terms.

## 2.5 Evaluating model fit and behaviour

The statistical models generated here can be viewed both as a GLM and as a simulator of fire occurrence for use in an ESM context, and as such can be evaluated through these two lenses. As GLM we evaluated the models using deviance explained; we made partial response and residual plots for each predictor on the link scale as is typically done for such analyses using 290 the R *visreg* package (Breheny and Burchett 2017); and we calculated variable importance using a SHAP-derived variable importance ranking using the R *vip* package (Greenwell and Boehmke, 2020). Note that because we didn't scale the predictors and did choose to use interaction and polynomial terms, we avoided the use of predictor coefficients or t-statistics to compare variable importance. Overall, this form of evaluation looks at the model's ability to predict burnt fraction, with equal weight



given to all gridcells in the training dataset (regardless of how much of an LCT is present in the gridcell or what time of year
it is) and does not consider details of the spatiotemporal patterns.

We also undertook complementary evaluation of predicted burnt area using methods that are more typically used in an
ESM/DGVM context. We plotted the spatial, interannual and seasonal patterns of burnt area and compared them to the
observations. We also calculated the normalised mean error (NME) of the spatial and interannual burnt area distribution and
the mean phase difference (MPD) to quantify the seasonal agreement, all following (Kelley et al.,
2013), compared to the data (over the full datasets, not just the training or testing subsample). We also plotted the predictor
responses on the response scale (with all other predictors are held at their median values) and compared them between the
LCTs to give a more "real world" idea of how these predictors act. It should be noted that these evaluations were done with
burnt area as opposed to burnt fraction, so when the model predictions are aggregated for the spatial temporal plots they
implicitly weight the gridcells' contribution by the fraction of LCT present. This implies that gridcells with less of a LCT
present contribute less to the plot.






**Figure 1. Breakdown of burnt area per land cover types and spatiotemporal patterns of cropland vs NCV burning.**





**3 Results**

**3.1 Analysis of observed fire occurrence by land cover type**

Of the 1.46 Mha/year of burned area in our European study domain between 2001 and 2020, the majority occurred in croplands, with a mean of 1.09 Mha/year (74% of total) (Fig 1a). The majority of this was in herbaceous croplands (0.98 Mha/year), with a much smaller contribution from mosaic croplands (0.08 Mha/year) and a very small amount in woody croplands (0.01

Mha/year). Given that cropland burning comprised three quarters of total burned area we conclude that our study should include cropland burning. Furthermore, since 90% of this burning happened in herbaceous cropland we decided to neglect burning in the other crop land cover types and consider only herbaceous cropland with the expectation that this corresponds to the practice of burning crop residues.

A further 0.38 Mha/year (26% of total burnt area) burned in non-cropland vegetation (Fig 1a). The largest contributions to this were from the Woodlands and Natural Mosaics categories (0.13 and 0.12 Mha/year respectively), with smaller contributions from Grasslands (0.06 Mha/year), and Shrublands (0.03 Mha/year), and a negligible contribution from Sparse Vegetation (0.003 Mha/year).

Although both NCV and cropland burning are generally confined to southern Europe, the spatial patterns of burnt fraction were rather distinct (Fig. 1b). Cropland burning was concentrated in the Balkans, particularly in Bulgaria and Romania on their respective sides of the Danube and around northern Serbia, and with some further patches in Italy and less intense patches in the Iberian peninsula. In contrast, NCV burning was most intense in Portugal, with further patches across much of the Mediterranean including particular hotspots in Sicily, the Balkan Adriatic coast and northern Serbia. Comparing the

normalised time series of burnt fraction of cropland and NCV revealed very different IAV (Fig. 1c). The seasonal distribution of burnt fraction (Fig. 1d) also showed some considerable differences, with a broader summer peak and distinctive October shoulder in the cropland burning. Despite the fact that NCV and cropland burning showed very different spatiotemporal patterns, our analysis of the NCV subtypes (including grasslands, shrublands, and woodlands) revealed remarkably similar distributions (see Figs A3-A5). From this we concluded that it would be sufficient to build GLMs for only two broad land

cover classes to capture the fire patterns in Europe: herbaceous croplands (henceforth just "croplands") that experience residue burning and all other non-cropland vegetation (NCV) which primarily experience uncontrolled wildfires, which we refer to as *BASE cropland* and *BASE NCV*, respectively.





**Figure 2. Partial responses of burned area to each predictor variable for the NCV (green line and shaded area) and cropland (purple line and shaded area) burning models. Dashed lines for the NCV and cropland burning model mark the burnt area predicted by the BASE models when all predictors are held at their median values.**



## 3.2 BASE deviance explained and predictors

The final selection of predictors are shown in Table 1 and the associated regression coefficients in Tables C1 and C2. In terms of raw deviance explained, the fitted GLMs did moderately well, explaining 32.4% of deviance of NCV burning and 36.0%
of cropland burning (Tables 2 and 3).

For the BASE NCV, the SHAP-derived importance scores indicated that the most important driver was FWI (log-transformed), closely followed by MEPI (Fig. 2). These relationships were positive and negative, respectively, as would be expected (Fig. 3). HDI (negative response), tree cover (unimodal response), and FAPAR12 (positive response) formed a group each with
approximately one third of the importance, closely followed by topographic slope (positive response). TPI and Pop_dens were of small importance and both had a positive response. Additionally, we found that including an interaction between FWI and MEPI had a small beneficial effect on the reproduction of the IAV and seasonal cycle, see Appendix E for details.

In BASE Cropland, MEPI was the most important determinant variable, with a negative response best represented in quadratic
form (Figs. 2 and 3). PHI (positive), FWI (unimodal), GDP (negative), and GPP12 (unimodal) all showed high and similar levels of importance (about two thirds that of MEPI) and wind speed, population density and slope showed less importance (all negative). We further note that many predictors produced contrasting responses and different functional forms in the NCV versus cropland model (Fig. 2).

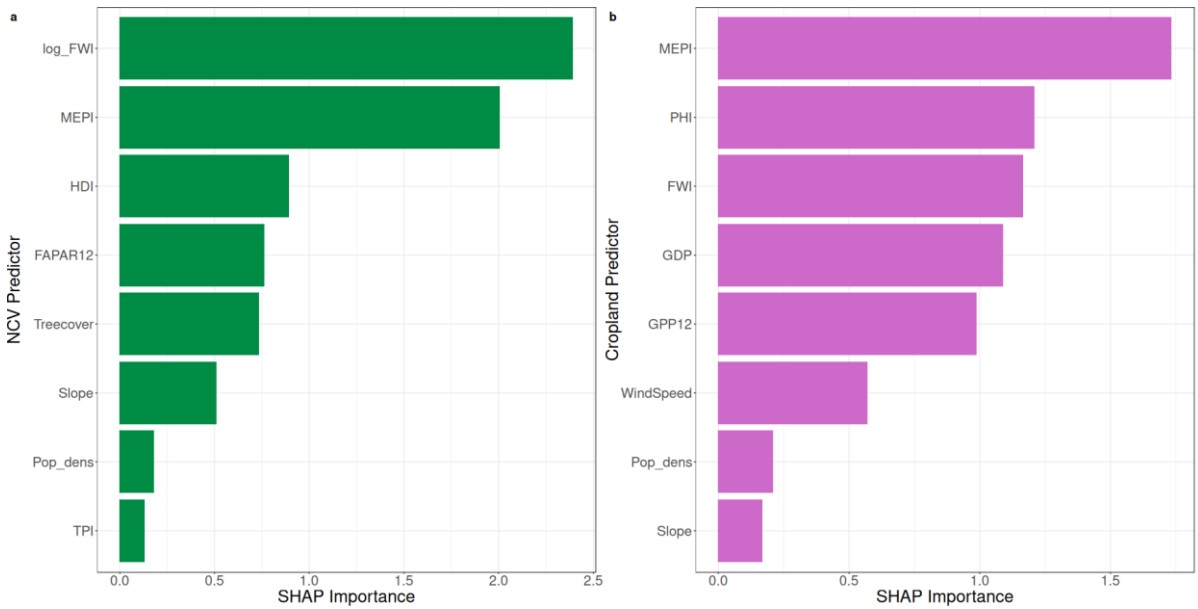


**Figure 3. SHAP-derived variable importance for a) the NCV and b) the cropland models. For variable descriptions see Table 1.**





| Description | Deviance explained | Spatial NME (1.066) | MPD (0.409) | IAV NME (1.235) |
|---|---|---|---|---|
| **BASE v1.0** | 0.324 | 0.867 | 0.284 | 0.581 |
| Omit FWI | **-0.206** | -0.003 | **0.110** | **0.357** |
| Omit HDI | **-0.042** | **0.067** | -0.004 | **0.084** |
| Omit Treecover | **-0.013** | **0.045** | -0.002 | **0.009** |
| Omit FAPAR12 | **-0.010** | **0.011** | -0.002 | *-0.005* |
| Omit MEPI | **-0.050** | *-0.008* | -0.001 | **0.139** |
| Omit Pop_dens | -0.002 | **0.006** | 0.000 | **0.013** |
| Omit Slope | **-0.011** | *-0.015* | -0.001 | **0.021** |
| Omit TPI | -0.001 | **0.005** | 0.000 | **0.006** |
| Include wind speed | 0.000 | -0.001 | 0.000 | -0.001 |
| FWI not logged | **-0.045** | **0.024** | **0.034** | **0.160** |
| MEPI and FWI not interacting | -0.002 | -0.002 | -0.002 | **0.011** |
| Pop dens quadratic | 0.000 | -0.001 | 0.000 | -0.001 |
| MEPI quadratic | -0.001 | -0.001 | *-0.006* | **0.017** |
| Treecover not quadratic | **-0.007** | **0.015** | -0.001 | **0.017** |
| Replace FAPAR12 with GPP12 | **-0.009** | **0.013** | -0.002 | *-0.019* |
| Include HDI x Pop_dens | 0.000 | -0.001 | 0.000 | -0.001 |
| Replace HDI with GDP | **-0.017** | **0.032** | -0.002 | **0.058** |
| Replace HDI with Pop_dens x GDP | **-0.017** | **0.032** | -0.002 | **0.058** |


**Table 2. Model skill metrics for best NCV burning model and the differences relative to the model for sensitivity models. Changes above 0.005 are highlighted in italics if they correspond to model improvement, and bold if they correspond to model degradation. The numbers in parenthesis in the column titles refer to the bootstrap null model.**




| Description | Deviance explained | Spatial NME (1.081) | MPD (0.338) | IAV (1.35) | NME |
|---|---|---|---|---|---|
| **BASE v1.0** | 0.360 | 0.610 | 0.258 | 0.988 | |
| Omit FWI | **-0.092** | **0.052** | **0.027** | **0.359** | |
| Omit GDP | **-0.055** | **0.096** | 0.002 | *-0.126* | |
| Omit GPP12 | **-0.016** | **0.016** | 0.000 | **0.027** | |
| Omit PHI | **-0.031** | **0.011** | *-0.026* | **0.087** | |
| Omit MEPI | **-0.072** | **0.045** | *-0.014* | **0.132** | |
| Omit Pop_dens | -0.002 | **0.005** | 0.000 | **0.010** | |
| Omit Slope | -0.002 | -0.001 | 0.001 | **0.021** | |
| Omit wind speed | **-0.008** | -0.003 | 0.000 | **0.039** | |
| Include TPI | 0.000 | 0.000 | 0.000 | -0.001 | |
| FWI not quadratic | **-0.026** | **0.029** | **0.009** | 0.001 | |
| GPP12 not quadratic | **-0.015** | **0.015** | 0.000 | **0.038** | |
| MEPI not quadratic | **-0.009** | **0.014** | 0.000 | **0.020** | |
| PHI quadratic | 0.001 | 0.000 | 0.000 | -0.003 | |
| MEPI PHI interacting | **-0.007** | **0.014** | -0.001 | **0.025** | |
| Replace MEPI with GPP | **-0.023** | **0.029** | 0.002 | *-0.021* | |
| Include GDP x Pop_dens | 0.001 | -0.004 | 0.000 | **0.012** | |
| Replace GDP with HDI | *0.006* | *-0.005* | 0.001 | **0.125** | |
| Replace GDP with Pop_dens x HDI | *0.007* | *-0.009* | 0.001 | **0.133** | |

**Table 3. Model skill metrics for best cropland burning model and the differences relative to the model for sensitivity models. Changes above 0.005 are highlighted in italics if they correspond to model improvement, and bold if they correspond to model degradation. The numbers in parenthesis in the column titles refer to the bootstrap null model.**





### 3.3 NCV model performance

The spatial patterns of burning simulated by BASE NCV matched the ESA FireCCI51 data reasonably well (Fig. 4) as
evidenced by a NME score of 0.87 (Table 2).  Burnt area occurred and was simulated largely in southern Europe in the Iberian
peninsula, the Balkans, Italy and Mediterranean islands.  However there were some regional mismatches.  The most striking
mismatch is the large overestimation by BASE in Spain and the simultaneous underestimation in Portugal.  BASE also
overestimated burning in Sardinia and Greece, but failed to simulate the high amount of burning along the Balkan Adriatic
coast.  Observational data also showed some areas of fire occurrence in temperate and boreal Europe (these may correspond
to a single fire event) which were not simulated by BASE.  The IAV was well captured (Fig. 5), with an NME of 0.58 and the
reproduction of both the observed weakly declining trend and the timing of peak fire years (although peak amplitudes were
underestimated).  The model also reproduced the observed timing of both the spring and summer fire peaks, but underestimated
their magnitude (Fig. 6), and produced an overall MPD of 0.28

### 3.4 Cropland model performance

BASE Cropland successfully simulated the large extents of cropland burning in the Balkans, Greece and Italy (Fig. 4) and
gave a good overall spatial NME of 0.61 (Table 3).  It did, however, considerably overestimate cropland burning across the
Iberian peninsula where little burning was present in the ESA FireCC51 data. In terms of interannual variability the model did
less well (Fig. 5), with an NME of 0.99. The model reasonably reproduced the observed seasonal timing of cropland burning,
with a MPD of 0.26, but underestimated the length of the Summer fire peak considerably and missed the Spring peak (Fig. 6).

### 3.5 Alternative model formations

Tables 2 and 3 show the performance of the fitted sensitivity models.  Changes equal to or larger than 0.005  (i.e 0.5% change
in deviance explained or NME) are in bold when they decreased model fit, and italics if they improved it.  In general, all
changes from the chosen model either worsened model agreement metrics or had negligible impact.  In the rare case that a
metric improved, it was almost always accompanied by a larger decrease in performance as measured with other metrics.

One noteworthy result is the large degradation of the IAV performance of BASE Cropland associated with the inclusion of
GDP, which resulted in a 13% decrease in IAV NME.  However, this was set against improvements in deviance explained and
spatial NME (6% and 10% respectively).  We also investigated swapping HDI for GDP, which resulted in a further 13%
degradation of IAV NME with only very small improvements in deviance explained and spatial NME.  Examination of the
temporal trends showed that HDI was responsible for introducing a decreasing trend in the cropland burning model which is
not observed in the data (Fig. E1), which explains its deleterious effect on IAV NME.  We also note that the inclusion of either
HDI or GDP in the cropland burning model improves the broad spatial patterns markedly by increasing cropland burning



the Balkans and decreasing it in western Europe (Fig. E2). We therefore chose to include GDP because of its lesser negative impact on the IAV NME. In contrast, in BASE NCV we found that the inclusion of HDI improved the description of the trend,

as the simulations correctly produced the slightly decreasing trend found in the data, as opposed to an incorrectly increasing trend without HDI included (Fig. E3). We also found that HDI was a superior predictor in all respects to GDP for NCV burning (Table 2).






**Figure 4. Spatial patterns of NCV and cropland burning for BASE and ESA FireCCI51 from years 2002 to 2014.**



**Figure 5. Annual time series of cropland (top panel) and NCV (bottom) burning for BASE and ESA FireCCI51.**





**Figure 6. Seasonal cycles of cropland (top) and NCV (bottom) burning for ESA FireCCI51 (solid line) and BASE (dashed line).**



## 4 Discussion

The results from the fitted GLMs broadly conformed to our expectations of the drivers of fire occurrence in both land cover types. The models demonstrated reasonable explanatory power when viewed as statistical models, and when viewed as simulators of fire occurrence, they showed similar model skill to global more complex vegetation-fire models applied at global

scope in terms of NME scores (Hantson et al., 2020). The models were constructed using predictors which are either easily calculable in most DGVMs or which can be taken as prescribed input layers (including pre-existing future projections where appropriate). We suggest that these models could be integrated immediately into DGVMs for application at European scale, particularly as we are not currently aware of any such models developed specifically for Europe.

Splitting the burnt area by LCT proved to be particularly illuminating. Whilst there was considerable overlap of the drivers of cropland and NCV burning, there were also some marked differences in terms of the direction of the responses and functional forms that provided the best fit. Furthermore, cropland burning modelling is comparatively underdeveloped; we are only aware of two global fire-enabled DGVMs that do so (Burton et al., 2019; Li et al., 2013) and one that prescribes it (Rabin et al., 2018), although see also recent developments from (Perkins et al., 2024). Recent studies have indicated that there is a

larger amount of cropland burning than previously estimated (Chen et al., 2023; Hall et al., 2024). Indeed, in this study cropland burning comprised 74% of the total burnt area, much higher than our initial expectation. Although these may not strongly impact the global carbon cycle, the trace gas emissions associated with this burning have significant implications for regional air quality and atmospheric chemistry, and also have ecological implications for cropland soils. The results and methods here can be tested and extended to simulate cropland burning at global scale.

### 4.1 Contrasting drivers of NCV and cropland fire occurrence


Cropland areas are fairly evenly distributed across Europe (Fig. A1) with the exception of north Scandinavia and the west coast of British isles where no fire occurs. We have demonstrated that within this broad European climate niche (Mediterranean, temperate, boreal), croplands burn with very different temporal and spatial patterns compared to NCV. Consistent with this observation and our expectations, we also found the drivers of each to differ in some respects (variables

associated with fire danger and spread, population density and vegetation properties), while retaining some broad similarity in others (MEPI, socioeconomic development).

#### 4.1.1 Fire weather and spread rate drivers

While NCV burning increased with FWI as would be expected, cropland burning showed a unimodal peak at intermediate FWI values. This makes sense as farmers would likely not burn during the most intensive period of fire weather due to the

risk of losing control of the fire or contravening fire bans.





Similarly, wind was not found to be a useful predictor in BASE NCV (likely because of the monthly resolution and because local wind speeds are highly modified by terrain and vegetation cover), but was a negative predictor for BASE Cropland. Topographic slope was a positive driver in BASE NCV but a negative one for BASE Cropland. Again this can be understood
as farmers not burning in circumstances which will encourage fast or unpredictable fire spread - i.e. during windy periods and on steep terrain.

### 4.1.2 Population density

We also saw opposing effects of population density on burned area in each LCT. For NCV, our results show that more people imply more fire, which is consistent with the logic that more people and infrastructure cause more fire starts (Haas et al., 2022).
But for croplands, we found that more people imply less fire, perhaps because burning near population centres is forbidden, or because burning is forbidden generally and only enforced near population centres, or burning is unpopular with local residents.

### 4.1.3 Vegetation properties

We also found the vegetation-related drivers of fire were different between the different land cover types. FAPAR during the preceding 12 months, a proxy for fuel buildup (particularly fine fuels), has a positive effect in BASE NCV, in line with a
previous study (Kuhn-Régnier et al., 2021). However, in BASE Cropland the best similar predictor was GPP of the last 12 months with a quadratic form which gave a strong response at intermediate values. The low burning at low values of GPP can be easily explained by insufficient biomass to burn. The low burning at high GPP is harder to explain, but we suggest this may be because higher GPP areas have more intense farming practices which use less fire.

For NCV burning, we found that intermediate levels of tree cover had the strongest positive effects on fire activity, although the exact mechanisms behind this correlation are hard to attribute. We suggest that this might occur because semi-forested ecosystems are, on the one hand, productive enough to produce sufficient quantities of fuel and, on the other hand, open enough that a lot of this fuel will be surface fine fuels - grasses and shrubs - which strongly support fire spread. This openness also implies a drier and windier microclimate, which will also encourage fire spread. Further work is required to disentangle these
mechanisms and we further note that some caution is required here as it cannot be excluded that the reduced tree cover in such areas is a consequence of fire occurrence rather than a cause of it.

### 4.2 Simulation of the seasonal cycle and the monthly GPP-derived indices

From very few monthly predictors BASE produces an acceptable representation of the seasonal fire cycle. In particular, BASE NCV predicts seasonal fire patterns very well using only two monthly predictors (MEPI and FWI) and their interaction
(technically FAPAR12 is monthly but its effect is heavily damped because it is a 12-month rolling mean). We suggest this offers a simple approach for separately quantifying two distinct but easily-conflated factors when considering fire danger: 1) the meteorologically-determined fire weather risk, here captured by FWI, and 2) the moisture and phenological status of the



vegetation, captured here using MEPI. The good performance indicates that using FWI offers a simpler alternative to using multiple climate variables as has been used in other studies. These climate variables are often highly correlated, which may
risk overfitting and cause difficulties for causal attribution. Similarly MEPI - being based only on GPP - provides a means to use current vegetation functioning to determine vegetation state, and therefore flammability, in a single predictor variable.

The seasonal cycle produced by BASE Cropland does not match the observations as well those produced by BASE NCV despite BASE Cropland using a larger number of monthly predictor variables. The factors determining the timing of cropland
burning are less clear than for NCV burning as its timing is controlled by human agency rather than biophysical factors. Flammability must still play a role in cropland burning, and our results indicate that preferred residue burning conditions are intermediate fire weather and low wind speeds. Beyond this it is a matter of agricultural practice and crop rotations which are not taken into account here but could be explored in future work with an expanded version of BASE.

### 4.2.1 Applicability of MEPI and PHI

The monthly ecosystem productivity index (MEPI) was constructed for this study as a robust way to quantify months when an ecosystem is in a more flammable state - which we equate to be when it is not photosynthesizing, either because it is phenologically dormant or under drought stress. We also note that over croplands, relatively low GPP may also occur after harvest if photosynthesising biomass has been removed or at the end of the summer, and it is at these times when residue clearing fires may occur. This approach worked well, and MEPI gave the expected response and was the most and second-
most important variable in BASE Cropland and BASE NCV, respectively. However, MEPI also includes periods when photosynthesis is low due to other factors which does not necessarily imply high flammability, in particular because of low temperatures. This will always be mitigated to some extent because low temperatures imply low fire risk, but it may be possible to improve the index to better handle periods where low temperatures reduce photosynthesis.

We designed the post-harvest index (PHI) to identify preceding three-month periods with high GPP as an indication that crops could have been grown to the point of harvest, and therefore when burning to clear residues may occur. This was to keep maximum model flexibility by avoiding the use of data specific harvest dates which might not be available. PHI was a high-importance predictor which improved the overall deviance explained and the spatial and interannual patterns, although it decreased the model skill with respect to seasonal patterns. This may be because the indicator was designed to capture fire
immediately after summer harvest, but may not predict spring crop burning.

Overall, our results suggest that defining indicators based on monthly GPP is a promising approach. We chose GPP over greenness measures such as NDVI because it can capture the ecosystem response to hot dry conditions (i.e. reduced photosynthesis due to water stress) before changes in greenness occur. Furthermore, GPP is a standard variable in DGVMs
with recent advances in both observing it using solar-induced chlorophyll fluorescence (Mohammed et al., 2019) and





simulating it using eco-evolutionary optimality methods (EEO, Stocker et al., 2020; Wang et al., 2017a). Thus GPP and derived indicators could provide a relatively robust contact point for coupling DGVMs and fire models.

**4.3 Strengths and weaknesses of BASE**

BASE NCV did a good job of reproducing the timing of spatially-aggregated seasonal and interannual fire observations, but
it underestimated both the seasonal and the long-term peak burned area amplitudes. It was less skillful in reproducing the observed spatial patterning of fire activity, which, although broadly correct, was too diffuse. The observed fire hotspots of central Portugal and the Adriatic coasts of Croatia, Montenegro and Albania were not reproduced, and in general the model failed to pick out local regions of burnt area. To some extent this may be because the data comprise discrete fire events which include a strong stochastic aspect that is inherently difficult for a statistical model (which predicts mean values) to reproduce.
However, even allowing for this, the details of the fire patterns in much of fire-prone southern Europe were not well captured. This might indicate an over-reliance on fire weather as a driver, along with a failure to include local factors that may lead to high danger such as particularly flammable vegetation types or high ignition risks due to human activities, land cover interface zones and infrastructure (Rodrigues et al., 2014).

These findings suggest that BASE NCV successfully captures broad fire drivers in terms of meteorological danger, coarse vegetation properties, and socioeconomic indicators, and is suitable for projecting future fire occurrence across Europe. Although the simulated spatial distribution of fire is imperfect, future work may improve this by including more detailed datasets concerning infrastructure, socioeconomic indicators, and vegetation types. However, such datasets might not be available for future scenarios and so including would inhibit the use of the model for future projections and so was not done
here.

In contrast to the BASE NCV, BASE Cropland's representation of the spatial distribution of fire occurrence is comparatively better than its temporal distribution. It picks out all the hotspots of cropland burning, although it does overpredict in some other regions. However, its simulation of the summer burning peak is too narrow and does not resolve the October shoulder,
while the interannual variability is poorly reproduced. It does capture a weakly increasing interannual trend, but this trend may be spurious as over a longer period cropland burning actually shows a decreasing trend (Fig. 1c). From this we can conclude we have captured the broad dependency of European cropland fires on socioeconomic development and suitable burning weather, but have not captured some specific factors affecting the likelihood or timing of burning such as sowing and harvest dates, crops types and systems (including double cropping systems), or legislative pressures, which would require
significantly more data input than is available at present in spatially gridded format.





### 4.3.1 Spain: an outlier demonstrating the importance of regional effects

In contrast to other southern European countries, Spain stands out for its low observed wildfire incidence, despite its fire-enabling Mediterranean characteristics. Here, BASE substantially overestimates both cropland and NCV fire occurrence, which may indicate phenomena specific to Spain which are not accounted for in BASE. We suggest that the answer may lie in changes made to its approach to wildfire risk at political and management levels during our study period. The period from 2003 to 2014 saw a decreasing trend in forest fires (Jiménez-Ruano et al., 2017; Vilar et al., 2015) due to the development, implementation and efficacy of wildfire suppression practices after the devolution of responsibility for them to regional authorities (Galiana et al., 2013; Pastor et al., 2020). This was associated with a decrease in the number of fire incidents and burned area due to improved fire control, potentially pushing fire sizes below the level which are detectable in the FireCCI51 product. The Spanish example highlights the importance of governance factors which cannot easily be quantified by broad indicators such as HDI or GDP, as well as the challenges associated with simulating fire regimes across multiple governmental or organisational jurisdictions. Inclusion of regional datasets and random effect terms (based on e.g. administrative areas or legislative changes) may improve model skill, increase understanding and be useful for short term forecasting. However, these techniques will be difficult to apply at larger spatial scales and in longer range projections, so accounting for such effects remains an open challenge for fire modelling at continental to global scales.

### 4.4 HDI and GDP as predictors of fire occurrence

Whereas GDP represents economic development, HDI reflects broad trends in human wellbeing across health, education and economic development. As such, neither explicitly captures the effectiveness of human fire management nor the tendency to utilise fire as a land management tool, and may be collinear with urbanisation and other infrastructural developments that may fragment landscapes and lead to declining burned area (Haas et al., 2022). Nevertheless, increasing societal wellbeing is likely reflective of increased state capacity for fire management and public awareness as well as the enforcement of environmentally-focused policies (Bhuvaneshwari et al., 2019; Zhang et al., 2020), and one could conjecture that HDI, as the broader indicator, will be a better proxy for such developments. Our results support this supposition, but only for NCV burning. We found HDI to be an important predictor for NCV burning; it was superior to GDP and its inclusion was important to capture the declining trend in NCV burning.

The picture is less clear for cropland burning. Both HDI and GDP improved the deviance explained and spatial patterns of cropland burning. We chose GDP over HDI for BASE Cropland because HDI introduced a declining trend in the cropland during predictions which is not seen in the data and correspondingly worsened the temporal NME. GDP did not introduce this trend and had less of a negative impact on the reproduction of IAV. GDP may be a better predictor because it more directly reflects capitalization and investment, which in turn directly affects agricultural practices and hence cropland burning.



However, we do not consider the result that GDP is a better predictor than HDI for cropland burning to be fully robust for a number of reasons. Firstly, due to the short record length and high IAV, the increasing trend in observed cropland burning (which appears to indicate that GDP is the better predictor) may be spurious. Indeed, a *decreasing* trend is seen over a longer period (Fig 1c.). Secondly, it is not clear what actually drives the IAV of cropland burning and *a priori* one would not expect such large variability, particularly as small fires generally have a lower IAV than larger ones (Randerson et al., 2012). Year-to-year variability in appropriate burning conditions will likely play a role and BASE Cropland clearly indicates that burning conditions are important. There are likely further influences on cropland burning IAV not captured here, possibly related to legislative factors, such as Bulgaria's and Romania's accession to the EU in 2007 (although we note that the decline in cropland burning commenced after 2007). However one would assume these would affect the trend rather than the IAV in cropland burning. Thirdly, it is possible that changes in land use, specifically the abandonment of croplands, are not immediately captured in the land cover data leading to the erroneous allocation of fires as cropland when they are in fact NCV. Such abandoned areas are considered as having high fire danger (Moreira et al., 2001; San-Miguel-Ayanz et al., 2012) which exacerbates the misclassification of land cover, and so wildfires in these areas may contaminate the cropland burning signal, with implications for IAV. Finally, inclusion of either GDP *or* HDI worsens the IAV NME (but GDP less so) whilst improving other metrics. Because these indicators represent broad socioeconomic changes through the years, we had expected them to improve the temporal reproduction of observations. We note that there is greater variation of HDI and GDP in space than in time, and it is likely that the spatial variation dominates the fitted model response. If so, this would imply that the temporal response of cropland burning to HDI/GDP is slower than would be expected based on the spatial gradients and that these indicators should be used with caution.

**4.5 Implications and outlook for simulation of crop residue burning**

The drivers of human fire use and their relation to wider fire regimes remain poorly understood at large spatial scales (Ford et al., 2021; Shuman et al., 2022). In our study area, cropland burning accounts for nearly three quarters of burned area, but has been comparatively understudied. For example, within the Database of Anthropogenic Fire Impacts - a global meta-analysis of academic literature on human-fire interactions - 43% (n=300) of instances of human fire use in our study area document prescribed fire to tackle extreme wildfires, whilst another 35% (n=245) of cases focus on diagnosing the human sources of unmanaged wildfires. By contrast, just 5% (n=38) of instances of human fire use document crop residue burning (Millington et al., 2022).

This study contributes to filling this knowledge gap by demonstrating how socioeconomic and environmental processes have contrasting impacts on cropland and NCV burning. This finding is in broad alignment with local-scale empirical studies (Millington et al., 2022) and the few existing modelling studies at large spatial extents (Perkins et al., 2024). Most importantly, we found that the drivers of crop residue burning differ in key ways from NCV fire occurrence. This is not surprising as they are very different phenomena. In Europe, the majority of NCV fires can be characterised as undesirable blazes, either set





accidentally or burning out of control, for which high fire danger and fuel loads play a large role. In contrast, residue burning is a deliberate process where socioeconomic factors and the avoidance of burning at times of high fire danger are highly relevant. These facts are codified in our results, providing tangible evidence that cropland and NCV fire must be modelled separately. Many previous attempts to reproduce large-scale burnt area patterns have not explicitly taken this into account, including many global fire models in DGVMs. We suggest that in the future better results can be obtained by explicitly simulating cropland fires and accounting for their different drivers and dynamics, or by simply removing burnt areas occurring in croplands from the target dataset if they are deemed not relevant.

This study elucidates some of the controls on cropland burning but further research is needed, particularly focussing on its temporal dynamics. For maximum flexibility and parsimony, BASE Cropland does not use information about harvest dates, crop types, or cropping systems, but including these could potentially provide new insights and improve the representation of the seasonal cycle, which is one of the weaker aspects of BASE Cropland. Introducing regionally specific factors to account for legislative changes, such as when countries joined the EU in which residue burning is, in principle, forbidden, may also prove helpful, in particular for understanding the temporal evolution of cropland burning. It may also help resolve questions about the use of the broader socioeconomic predictors such as HDI and GDP.

### 4.6 Limitations and caveats

### 4.6.1 GLM approach

Overall, our approach of fitting GLMs to monthly data worked well, and so demonstrates that this method can give serviceable fire occurrence estimates with a seasonal cycle suitable for integration into other models such as DGVMs. However, the approach does come with a few limitations. In common with other GLM studies (e.g. Bistinas et al., 2014; Haas et al., 2022), our model tends to "smear out" the burnt area by underestimating extremes and predicting many small values instead of zero. This may in part be due to the fact that in reality fire manifests in discrete events, whereas GLMs only predict mean values. This also explains the distinctive "many small underestimates, a few large overestimates" pattern in the partial residuals (Figs. D1 and D2). Another caveat is that due to spatial autocorrelation in the datasets, the standard errors are likely to be considerable underestimations and the uncertainty bands in Figs. 2, D1 and D2 should be viewed as lower bounds (although this does not affect the central parameter estimates). Finally, the necessary use of the quasibinomial distribution to model burnt fraction precludes the use of standard statistical tools and diagnostics such as QQ plots and information criteria. It may be possible to overcome some of these limitations in the future by moving away from burnt fraction as the target variable and focussing on other aspects of the fire regime, thus putting understanding of fire occurrence on a more rigorous statistical footing.



### 4.6.2 Uncertainties and errors in remote sensing data

This study relies heavily on remote sensing data, particularly the burnt area and land cover data used to construct the target variable. Remote sensing products based on MODIS (including the ESA FireCCI data used here) are known to struggle with detecting small fires and have high omissions errors, particularly in the Mediterranean (Katagis and Gitas, 2021). This has implications for all fires, but cropland fires in particular are typically small, brief, and low intensity, and more often missed by remote sensing (Hall et al., 2021; Zhu et al., 2017). So while it accounts for the majority of the burning in the study region, the estimate of cropland burning used here is likely an underestimation, and analyses with newer remote sensing datasets may modify and ultimately improve on our results. Misclassifications of land cover will also affect our results, in particular imperfect separation of croplands from other vegetation types due to rapid land use change (Winkler et al., 2021; Zubkova et al., 2023). Notably, abandoned croplands in their early stage of transition represent a major fire risk (Moreira et al., 2001; San-Miguel-Ayanz et al., 2012) and may cause commission error in the identification of cropland fires. However, the prevalence and predictability of this signal and the support from existing literature (Millington et al., 2022) gives us confidence that the bulk of this signal is indeed cropland residue burning. In particular, one recent study based on higher resolution remote sensing data in Romania (one of the cropland burning hotspots in Europe) confirmed many of the results here (Mattes et al., 2024), namely that the majority of burning in Romania is indeed in arable land; that these fires occur less in areas with steep topography; and that their frequency is reducing due to socioeconomic factors.

## 5 Conclusions

This study aimed to disentangle the drivers of fire occurrence across a European study domain and encapsulate this knowledge into a new fire model (the BASE model). Our initial investigations of burnt area in Europe revealed that cropland and non-crop vegetation (NCV) land cover types burn with very different spatiotemporal patterns. After fitting GLMs to each land cover type separately, we confirmed that there are very different drivers for fire occurrence in the land cover types. This was most clearly manifested in fire weather and other variables connected to rate of spread, where our results indicated that crop residue burning is preferentially conducted in situations of lower fire danger and rate of spread. This outcome has implications for any large scale studies to simulate burnt area over a mixture of land cover types. Our results also provide some novel insights into the drivers of cropland burning which has, to our knowledge, not previously been studied systematically over Europe. In addition to optimal burning conditions, we found a strong control on spatial patterns by socioeconomic development and on seasonal timing by GPP-derived indices. However, the mechanisms controlling the seasonality and interannual patterns of cropland burning remain poorly understood and require further study. This is of particular importance because fire occurrence in cropland has recently been revealed to be more prevalent than previously estimated.

Overall, the BASE model reasonably captures the spatiotemporal patterns of burnt area in Europe. BASE NCV does particularly well with reproducing observed temporal dynamics and relatively less well with the spatial patterns, while the

opposite is true for BASE Cropland. We suggest two potential future applications for BASE. As a simulator, BASE already

provides a serviceable means to simulate fire occurrence in Europe that is compatible and easily integrated with other model frameworks, such as DGVMs. In particular, its skillful reproduction of the seasonal and interannual patterns of wildfires indicate that it captures the temporal dynamics and so is suitable for projecting changes in fire hazard over annual-to-decadal time scales. The explicit simulations of cropland fires is also a noteworthy advance. Further to its use for projections, we suggest that the BASE framework may also be further utilised by considering additional potential predictor datasets to improve

understanding of the controls on burnt area in Europe. For BASE Cropland, data about harvest date and cropping systems, and variables to capture changes in legislation may help understand the temporal dynamics. For BASE NCV, additional socioeconomic indicators and maps of vegetation types and infrastructure may explain the spatial patterns. Both of these applications of BASE may help in meeting the challenges of increasing fire risk faced by Europe.

**Code availability**

Code used in this analysis (including data preparation, model fitting, analysis and plotting) are available at https://doi.org/10.5281/zenodo.12580481.

**Data availability**

The data used to fit the models are available are in a Zenodo repository at https://doi.org/10.5281/zenodo.12580343.

**Author contribution**

MF conceptualised the model and performed the data analysis and model fitting with support from MB, SB, JH, TH, EK, LO and KT. MB, SB, JH, EK and LO processed and provided datasets. OP provided context and interpretation for human dimensions of fire use and socioeconomic indicators. DW assisted with the development of the statistical framework. FAF provided context and interpretation for the southern european results. MF prepared the manuscript with contributions from all co-authors.

**Competing interests**

At least one of the (co-)authors is a member of the editorial board of Biogesciences.



**Acknowledgements**

This project has received funding from the European Union's Horizon 2020 research and innovation programme under grant agreement No 101003890 (FirEUrisk). From FirEUrisk, JH, MB, SB, and EK received salary and JH, MB, SB, EK, MF, LO, KT, and TH received travel support. In addition, FAF was supported by a predoctoral scholarship (FPI) from the Spanish Ministry of Science, Innovation and Universities (PRE2019-089208). OP is funded by the Leverhulme Centre for Wildfires, Environment and Society through the Leverhulme Trust, grant number RC-2018-023. LO is funded within the research training group "Natural Hazards and Risks in a Changing World" (NatRiskChange) funded by the Deutsche Forschungsgemeinschaft (DFG; GRK 2043/2). We thank Marcos Rodrigues for informative discussions.



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




# Appendices

## Appendix A: Additional land cover and fire occurrence per land cover plots

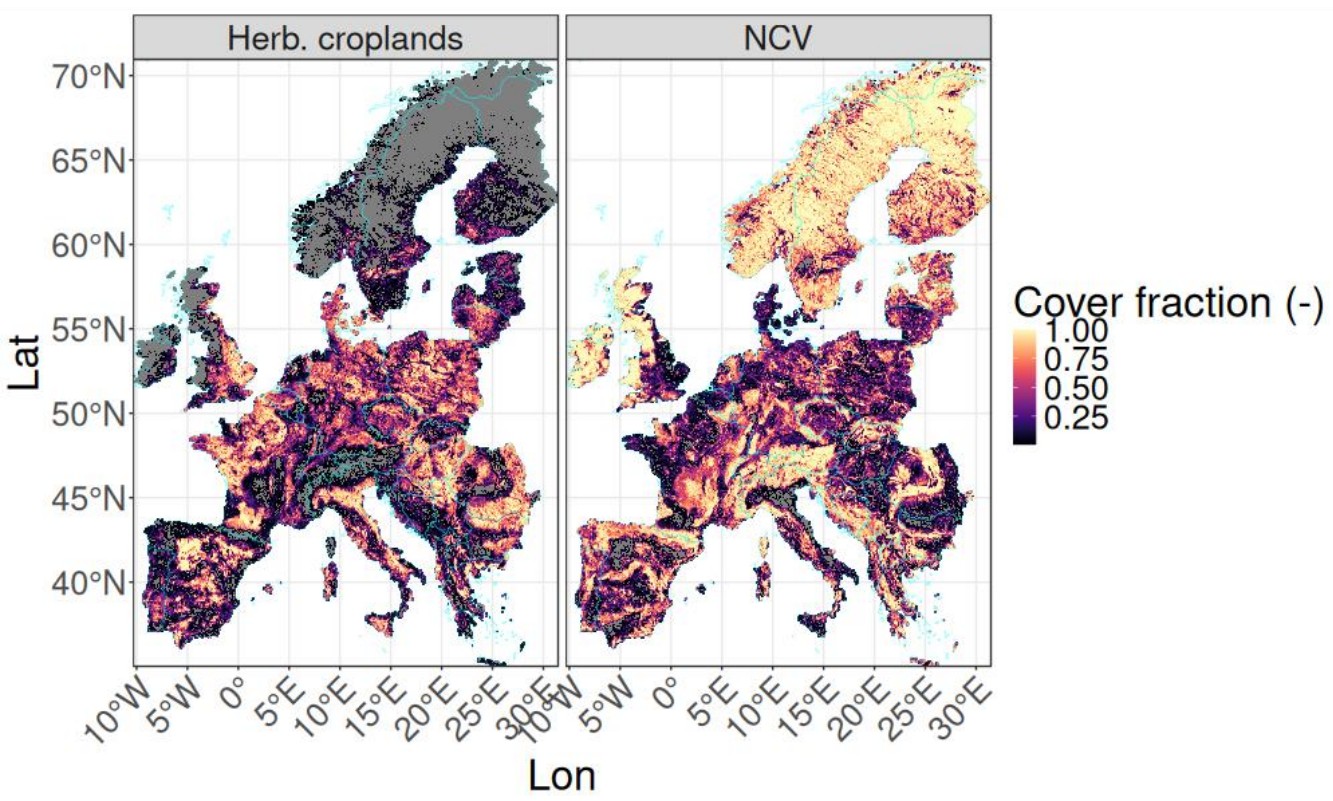


**Figure A1. Distribution of NCV and cropland land cover from ESA LandcoverCCI across the European study domain (average 2001-2020)**



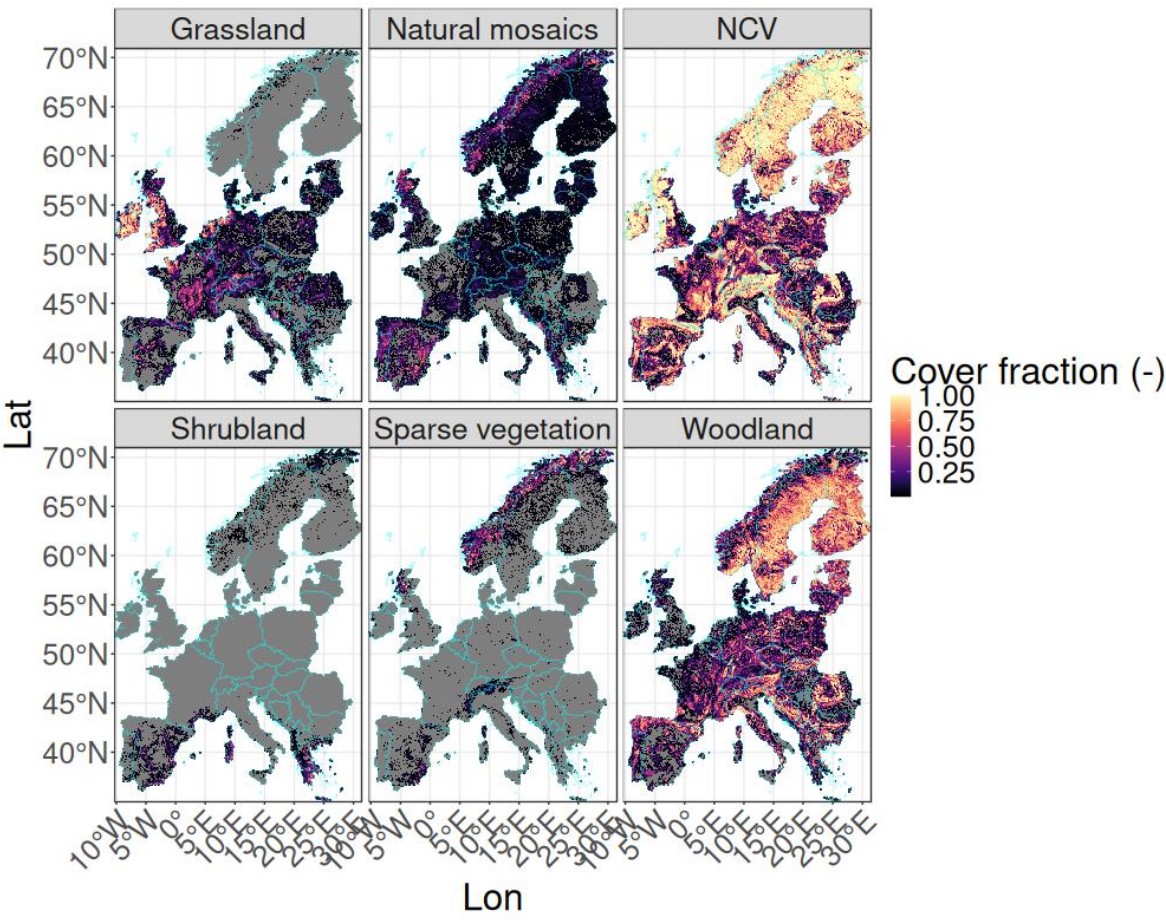


**Figure A2. Distribution of NCV land cover subtypes ESA LandcoverCCI across the European study domain (average 2001-2020)**





**Figure A3. Burnt fraction from ESA FireCCI51 in NCV land cover subtypes across the European study domain (average 2001-2020)**



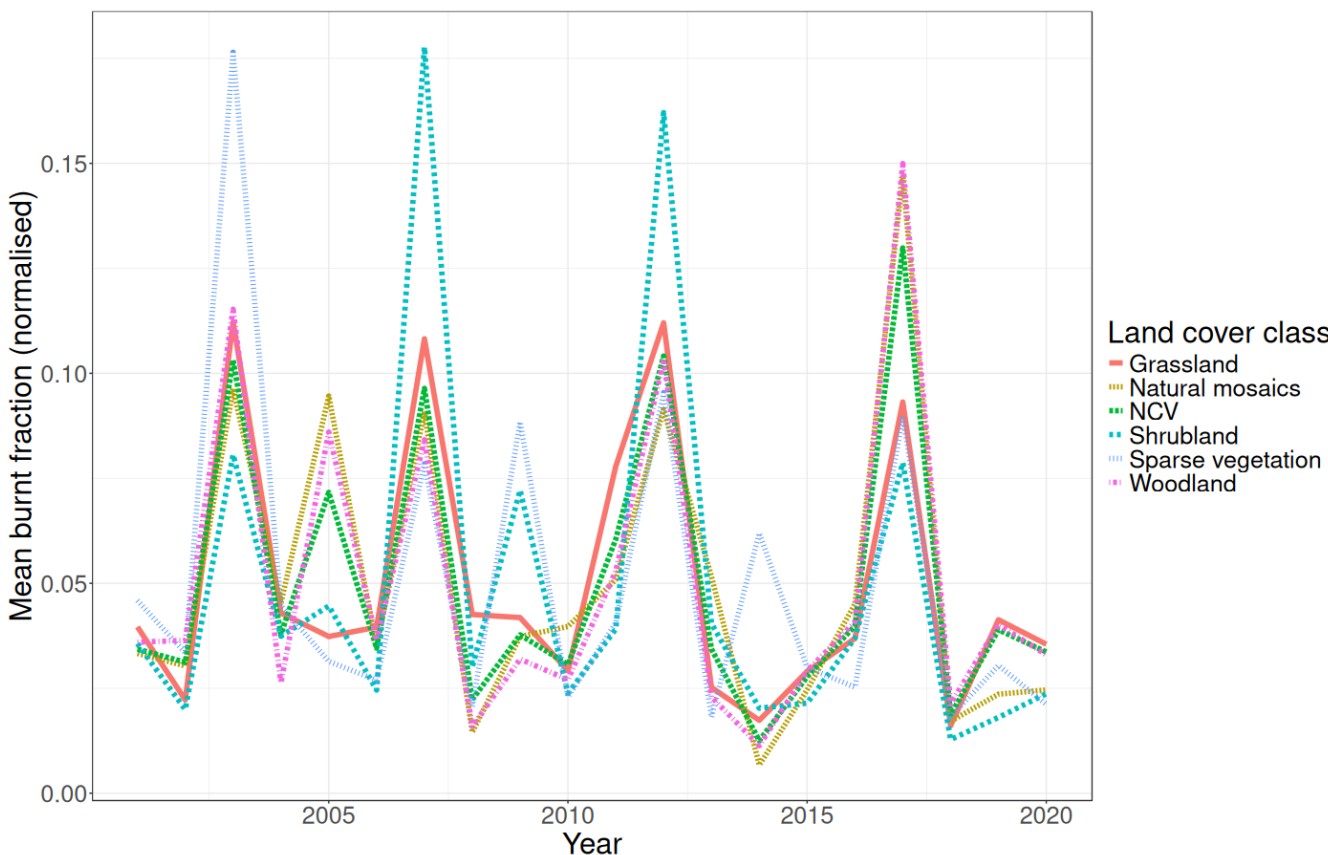

**Figure A4. Mean burnt fraction from ESA FireCCI51 in NCV land cover subtypes across the European study domain.**






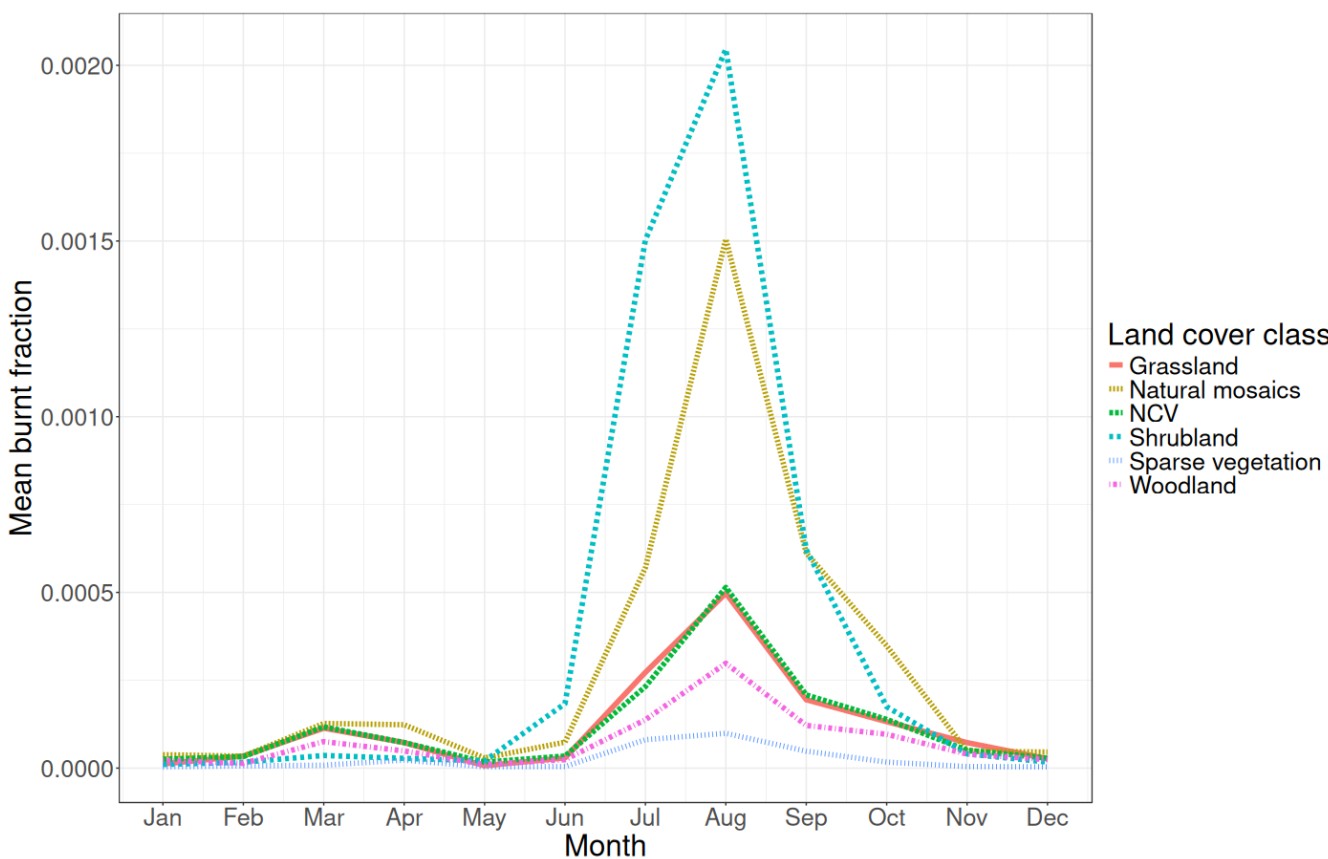

**Figure A5. Seasonal cycle of mean burnt fraction from ESA FireCCI51 in NCV land cover subtypes across the European study domain (average 2001-2020).**




Appendix B: Predictor correlations

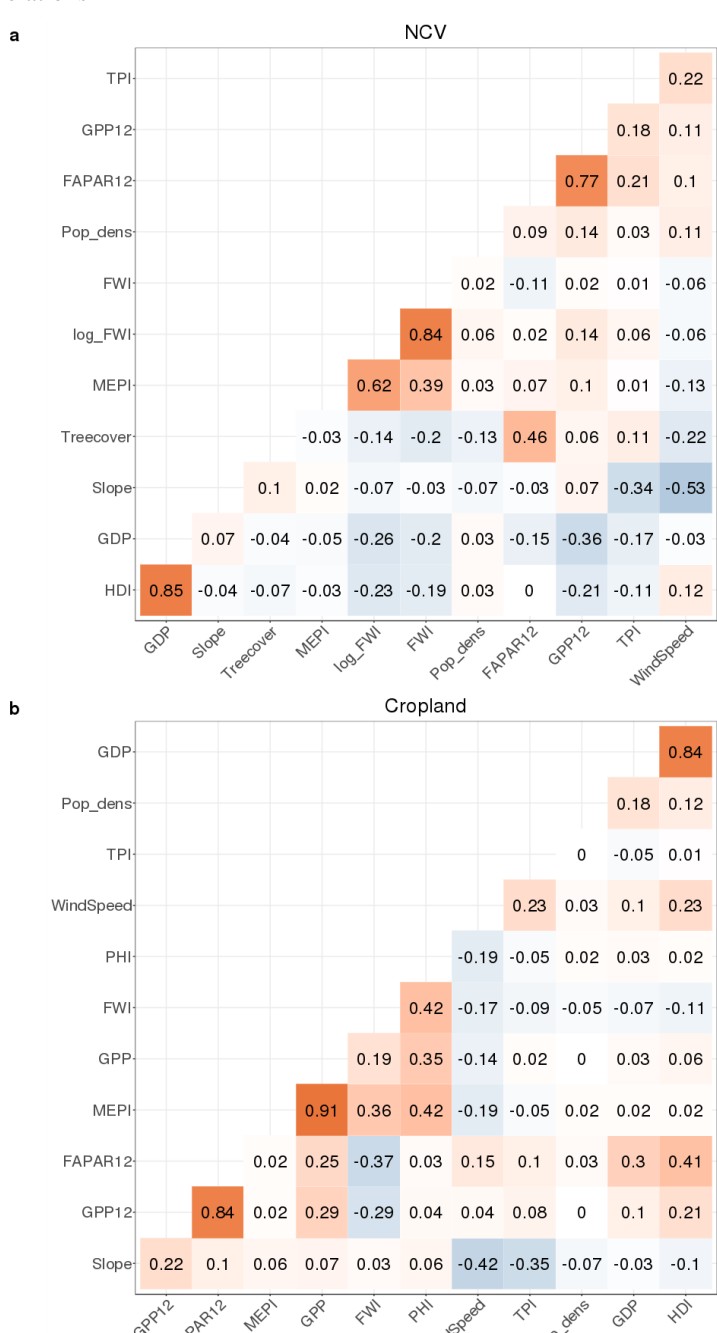

**Figure B1.** **Pearson's correlation of the predictors used in NCV and cropland BASE (final model and alternative model formulations).**





## Appendix C: Regression model parameters for the final BASE models

| Term | Value | Std error | t-statistic | p-value |
| --- | --- | --- | --- | --- |
| (Intercept) | -1.33000 | 0.274000 | -4.86 | 1.15e-06 |
| FAPAR12 | 6.07000 | 0.254000 | 23.90 | 0.00e+00 |
| Slope | 0.08890 | 0.003300 | 26.90 | 0.00e+00 |
| TPI | 0.48200 | 0.053600 | 8.98 | 0.00e+00 |
| Pop_dens | 0.02940 | 0.002680 | 11.00 | 0.00e+00 |
| HDI | -16.40000 | 0.331000 | -49.40 | 0.00e+00 |
| Treecover | 0.06990 | 0.005670 | 12.30 | 0.00e+00 |
| Treecover^2 | -0.00187 | 0.000103 | -18.20 | 0.00e+00 |
| MEPI | -6.76000 | 0.252000 | -26.80 | 0.00e+00 |
| log_FWI | 1.87000 | 0.032000 | 58.50 | 0.00e+00 |
| MEPI:log_FWI | 0.95300 | 0.081100 | 11.70 | 0.00e+00 |


**Table C1. Estimated coefficients, standard errors, t-values and p-values for the final NCV model. P-values below $10^{-16}$ are reported as zero.**

| Term | Value | Std error | t-statistic | p-value |
| --- | --- | --- | --- | --- |
| (Intercept) | -7.91e+00 | 1.53e-01 | -51.7 | 0 |
| Pop_dens | -3.26e-02 | 2.07e-03 | -15.8 | 0 |
| GDP | -2.67e-02 | 3.04e-04 | -87.9 | 0 |
| Slope | -5.28e-02 | 3.34e-03 | -15.8 | 0 |
| PHI | 3.70e+00 | 5.79e-02 | 63.9 | 0 |
| WindSpeed | -1.71e-01 | 4.86e-03 | -35.2 | 0 |
| FWI | 2.21e-01 | 2.46e-03 | 89.7 | 0 |
| FWI^2 | -2.93e-03 | 4.88e-05 | -59.9 | 0 |
| GPP12 | 9.54e-03 | 2.50e-04 | 38.2 | 0 |
| GPP12^2 | -4.58e-06 | 1.14e-07 | -40.3 | 0 |
| MEPI | -1.15e+01 | 1.96e-01 | -58.9 | 0 |
| MEPI^2 | 7.20e+00 | 1.84e-01 | 39.2 | 0 |

**Table C2. Estimated coefficients, standard errors, t-values and p-values for the final cropland model. P-values below $10^{-16}$ are**
**reported as zero.**





## Appendix D: Partial residual plots for the final BASE models

Figure D1. Plots of the partial residuals (orange to purple heatmap, note the logarithmic scale) and partial responses (cyan lines) for BASE NCV on the link scale. The black "+" symbols indicate the variables are involved in an interaction term the effect of which is not included here (see Appendix E).





**Figure D2. Plots of the partial residuals (orange to purple heatmap, note the logarithmic scale) and partial responses (cyan lines) for BASE cropland on the link scale.**






**Appendix E: Interaction terms**


When developing BASE we tested various interaction terms, however only one was retained in the final BASE configuration: the interaction between MEPI and log_FWI in the NCV model. Including this term improved the IAV NME by 1%, and had only had a very small impact on the other metrics (Table 2). It also improved the timing of the March and August peaks (Fig E1).


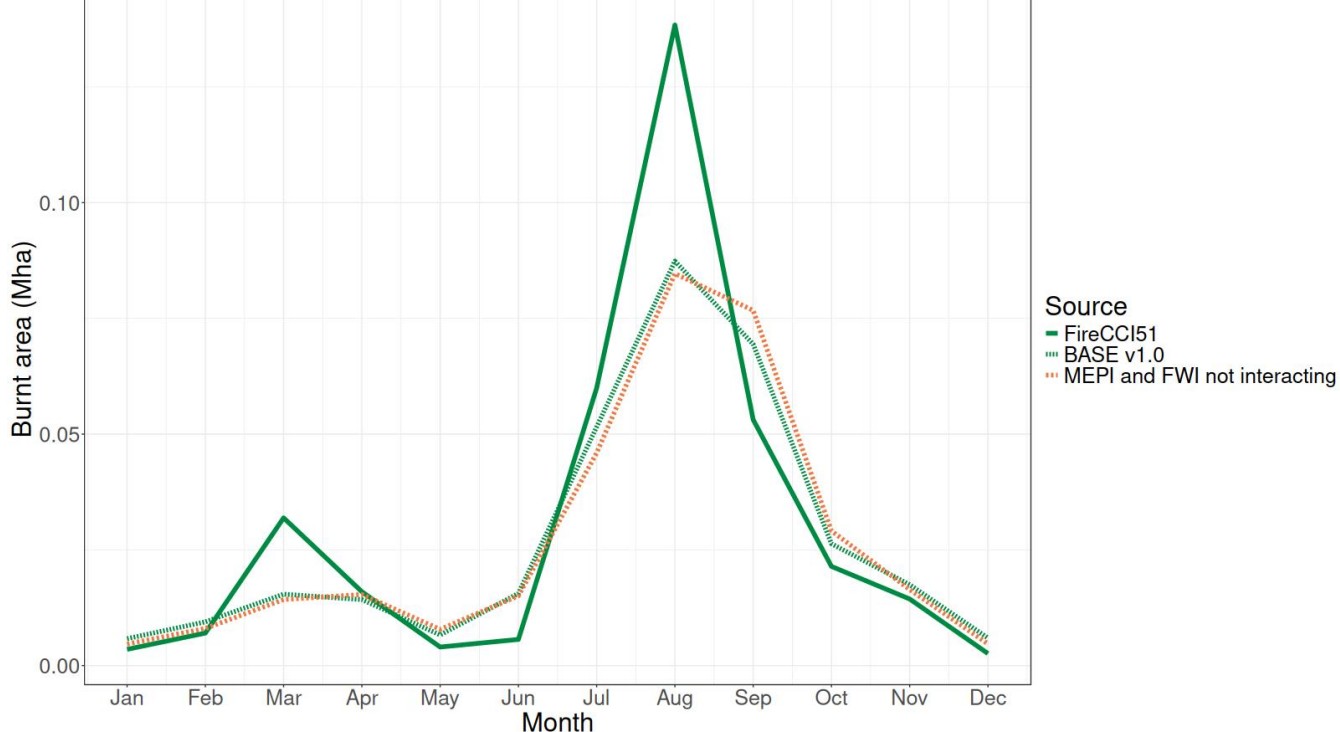

**Figure E1: Comparison of seasonal cycle of NCV burning in the final BASE configuration and the sensitivity model with the interaction between FWI and MEPI omitted.**


Visualisation of interaction terms requires special consideration as their effects cannot be included in typical 1D predictor response or partial regression plot. We took the approach of isolating the effect of the interaction terms, $\beta_{1,2}.x_1.x_2$, on the response scale and plotting that in two dimensions i.e. $(x_1,x_2)$ space. Similarly to the 1D plots, we kept the other predictors at their median values. In order to isolate an interaction term we first calculated the full model prediction on a 2D plane of $x_1$ and $x_2$ (analogous to a 1D response plot). We then calculated the response without the interaction term. Technically speaking, this was done by first, on the link scale, subtracting the $\beta_{1,2}.x_1.x_2$ interaction term from the full prediction. Note that this






prediction still includes the linear terms $\beta_1.x_1$ and $\beta_2.x_2$, so the interaction term $\beta_{1,2}.x_1.x_2$ is the only term that is removed. This was then converted to the response scale and subtracted from the full response, and then this difference was plotted to quantify the effect of the interaction term.


The contribution of this interaction term between MEPI and log_FWI as visualised by this method is shown in Fig 7. This indicates that the interaction increases the predicted burnt fraction at high log_FWI and low to intermediate MEPI.

### NCV interaction effect: MEPI*log_FWI




**Figure E2. Contribution of the MPEI x log_FWI interaction term on the response scale.**



It should be noted that although the interaction term must be monotonic in both dimensions on the link scale by its construction, the difference on the response scale will not necessarily be. This is because the inverse link function is not necessarily linear. In this case the inverse link function is the logistic function which is not linear and in fact plateaus towards an asymptote. This means that in some areas of the $(x_1, x_2)$ space ,the response is already very high without the interaction term (i.e on the plateau on the logistic function), and so adding the interaction term has very little effect on the response *even though the interaction term might be at its largest values on the link scale*. In other words, we don't necessarily expect the *interaction term on the link scale* and the *effect of the interaction term on the response scale* to have the same shape.




**Appendix F: Spatiotemporal plots for selected sensitivity models**

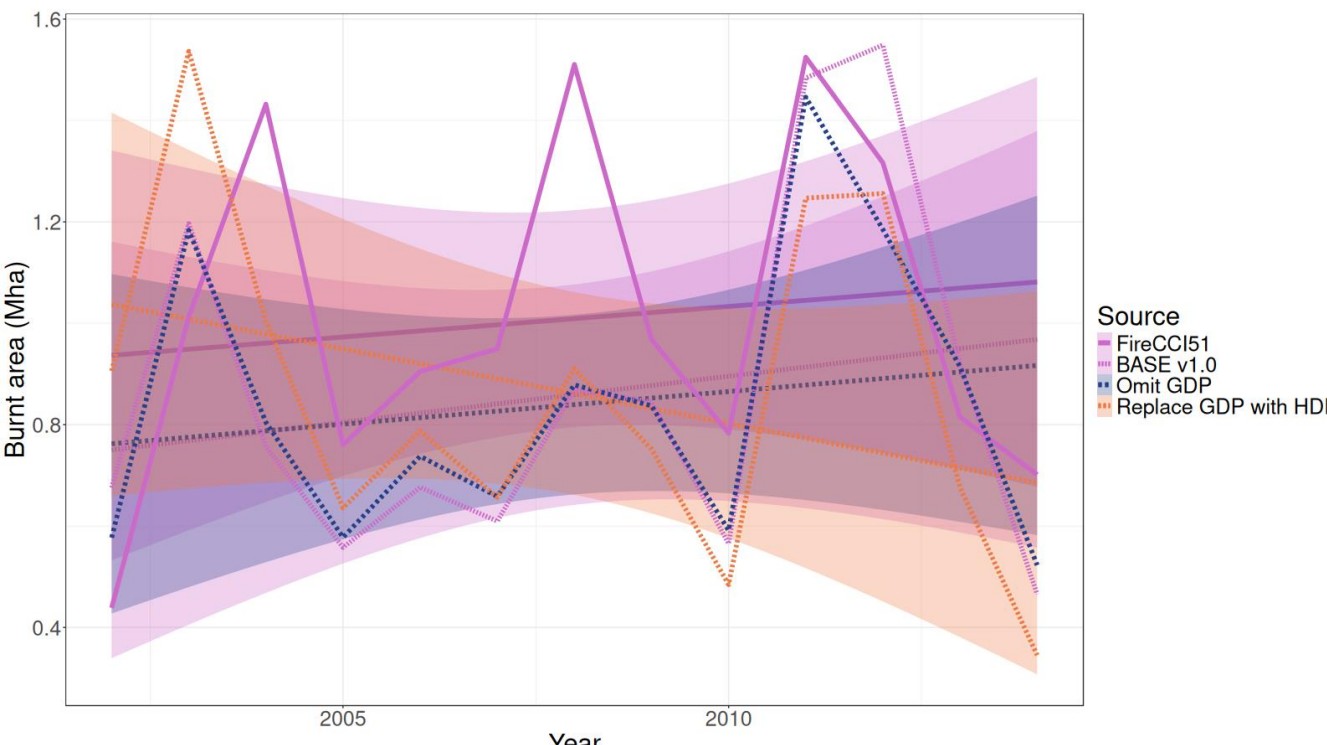

**Figure F1: Comparison of IAV of cropland burning in the final BASE configuration and sensitivity models with changed socioeconomic predictors.**





Figure F2: Comparison of spatial patterns of cropland burning in the final BASE configuration and the sensitivity models with changed socioeconomic predictors.



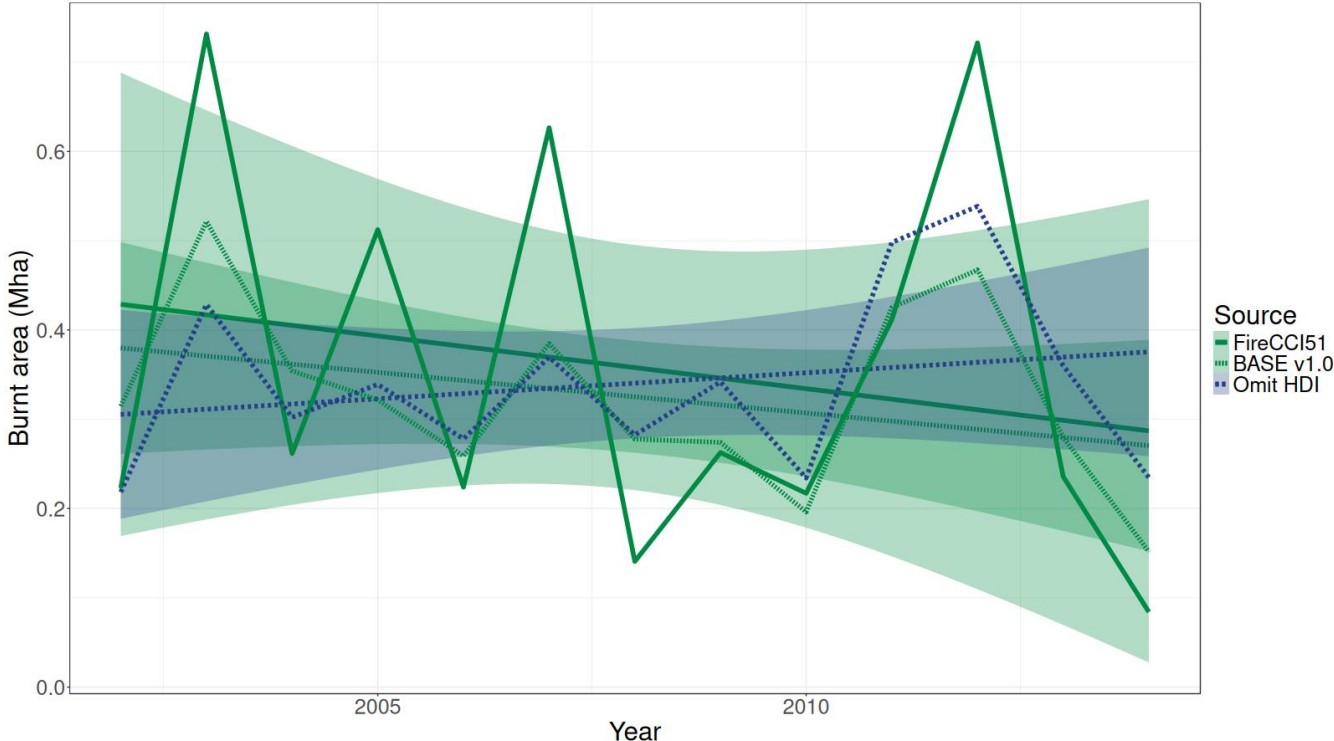

**Figure F3: Comparison of IAV of NCV burning in the final BASE configuration and the sensitivity model with HDI omitted**