# Peer review of "Understanding and simulating cropland and non-cropland burning in Europe using the BASE (Burnt Area Simulator for Europe) model"

_EGUsphere, 2024_

## Author Comment (AC1)

Dear Taimur Khan,

Thank you for the community comment on our manuscript submitted to Biogeosciences. We note that the comment does not contain any detailed discussion of fire, land cover or Europe and the comments are fairly generic. Sometimes the comments do not fit – for example the statement, "it could be strengthened by... how the BASE model could be refined or extended to improve its predictive power", we do in fact discuss exactly this – both for NCV and Cropland burning. Furthermore, the writing style looks very much like it was produced by a generative AI, and running the text through a couple of online tools suggests that this is indeed likely to be the case.

This is not to say that such comments automatically don't have merit (although they are no replacement for expert peer-review), so we will address them briefly here. The key points are strengthening the Discussion and clarifying methodological assumptions and uncertainties. With regards to the Discussion, it is already extensive and following some further suggestions from reviewers 1 and 2, we feel, complete. Because this work does not include future projections or apply at the landscape scale, we feel that the suggestion of "policy implications" is not appropriate here, although see Hetzer *et al.* 2024 for related work which include future projections of meteorological fire risk in Europe and policy implications.

As for methodological detail and uncertainties, we feel we have fully documented out methods and in fact already gone a little further the previous studies in the field in discussing these issues - in particular our discussion of spatial autocorrelations and their effect on estimated uncertainties (which motivated our decision to use Shapley variable importance instead of t-values), discussion of uncertainty in the satellite data, the consequences of using a *quasi* family in the GLM, and our discussion of interaction terms. We do not feel that there are any outstanding methodological issues to discuss.

References

 Jessica Hetzer *et al* 2024 *Environ. Res. Lett.* **19** 084017

---

## Author Comment (AC2)

**Response to RC1**

We thank the reviewer for taking the time to review our manuscript and provide thoughtful and interesting comments. These points often merit further discussion to which we have tried to do justice, but have tried to make minimal changes to the existing text as some of the discussions are a little speculative and getting deep into them would lengthen the (already rather long) manuscript.

In the following point-by-point response we have kept the reviewer's comment verbatim in black text, our responses are in blue and proposed new text in green. Line numbers refer to the reviewed version of the manuscript. Text insertions into existing text are additionally denoted with underlining.

The authors of "Understanding and simulating cropland and non-cropland burning in Europe using the Base model" use generalized linear models to develop a fire model capable of predicting cropland and non-cropland burned area in Europe. This model is likely suitable for use in land surface and climate models. To my knowledge, few land surface models include the ability to model cropland fire. This work is timely, technically rigorous, and falls within the scope of bio-geosciences. I have several comments which are listed below.

- Intro: The lack of land surface models capable of representing cropland fire is mentioned in the discussion. I suggest discussing it in the intro as well as the motivation for this work.

Yes, it was a motivation and we agree we should mention it more in the Introduction. Some land surface models do include cropland fire (CLM, since v4.5 but it is not enabled in all configurations used in CMIP6, and JULES-INFERNO, but not yet in the main JULES release) but these are a definite minority of models. We propose to add the following text after line 110:

"Cropland burning as an explicit process is almost entirely neglected in fire-enabled DGVMs and the land surface models used in Earth System Models (ESMs); we are aware of only one such model in which it is simulated (Li et al., 2013), one in which it is prescribed from remote sensing data (Rabin et al. 2018), and one in which fires in croplands are simulated in the same manner as fires in grasslands (Burton et al. 2019)."

- Table 1: Adding the data source and citations could help better inform the reader

Yes, good point, we are happy to do this.

- L270: How were the data points sampled? Was anything done to uniformly distribute the sampling across space, or account for spatial autocorrelation

They were sampled completely randomly in space and time (i.e not the complete time series for a gridcell, we also sampled from the individual months). We did try some degree of stratification (to balance the burnt for non-burnt gridcells) but that degraded out results. Initially we also tried sampling every alternate year, but the interannual fluctuation seem to alternate for at least part of the time series so that introduced a bias depending on whether we took every first or every second year. In the end we found that completely random sampling gave the best results. We propose to clarify by modifying the sentence at line 271 to read:

"We considered every month and gridcell which had more than 10% of the LCT present as a data point, and used 80% of the data points (sampled randomly from all grid cell-months) for training and kept 20% for testing"

- L460: Some text comparing and contrasting these models with mechanistic models could be interesting. For example in mechanistic models, wildland fire is influenced by wind and terrain which impacts spread, whereas cropland fire appears to be a more complex phenomenon perhaps better suited to description using a statistical model.

Yes, very good point that we missed. We agree that statistical (or agent based) modelling of fire definitely makes sense, at least until we know more about the processes. Therefore, we propose to include the following in a new paragraph at line 46 6.

"These results imply that current mechanistic modelling approaches are likely not well suited to modelling cropland fires. Mechanistic models are typically based on biophysical relationships concerning flammability and rate of spread, and with the general assumption that higher flammability or faster rates of spread produce more burnt area. Our findings imply that this approach is not valid for cropland burning as more flammable conditions do not necessarily imply more burning in the croplands. Given this, and the complexities of human land management and other socioeconomic factors, the inclusions of statistical or agent-based (Perkins et al. 2024) approaches in future cropland modelling efforts may prove fruitful."

- L575: The analysis of the role GDP and HDI play in the model is interesting. Can the authors provide insight into whether this is corelative or causative? Do these relationships apply in time (i.e. moving into the future)? What if there were abrupt changes in these metrics due to for example a short-term financial crisis?

Good questions. Essentially, they are correlative, but we argue that there is a strong "indirect causation" rather than "just correlation". It cannot be directly causative – higher GDP (or the other components of HDI which are years of schooling and life expectancy) does not directly affect fire occurrence. Rather, these variables are acting as a proxy for "socioeconomic development" in a broad sense, and, as this "development" occurs, they capture changes in human behaviour, infrastructure, legislation, etc which result in less burnt area. These "indirect causations" are not difficult to imagine. Higher GDP leads to more investment in fire-fighting capability and more capital-intensive, mechanised agriculture which doesn't involve using fire. More education leads to more awareness of air pollution and less tolerance of it. There are many ways that a more "developed" society stops using fire and actively suppressed it, and has the means and incentivise to do so. However, these are (as of yet) unquantified, so we don't wish to speculate too much in the manuscript about mechanisms, so we propose to clarify the causative nature of the relationships be modifying the sentence at line 578 to read:

"As such, they have correlative rather than causative relationships with burnt area as neither explicitly captures the effectiveness of human fire management nor the tendency to utilise fire as a land management tool, and may be collinear with urbanisation and other infrastructural developments that may fragment landscapes and lead to declining burned area (Haas et al., 2022)."

With regards to the relationships in time, yes, we believe the relationships do capture the effects of temporal changes in GDP/HDI as they are essential for capturing the declining trend in NCV burning (supp. Fig F3 in the manuscript). This is driven by a solid increase in HDI across the study region,

especially in regions such as the Balkans (although spatial differences are larger than the temporal ones). For moving to the future, we become sensitive to broad issues such as the universality of the trajectory of societal development and potentially more "developed states" than we have today (maybe where prescribed burning becomes commonplace or maybe suppression of extreme fires becomes even more effective). And also the specific caveat that applies to all models that they might not be reliable outside of the regime in which they are trained and tested. However, in practical terms for BASE and the state-of-the-art scenario modelling (i.e. SSPs), we believe that the model's response to GDP and HDI will likely to be reasonable, i.e. a general decrease in fire activity as the development indices increase (although there is a saturation at high values) and a difference in development between the SSP scenarios with different socioeconomic trajectories. This is apparent in some on-going work where we are looking at future projections.

Regarding abrupt or short-term changes. We are not aware of any studies relating changes in fire regime to abrupt socioeconomic changes, although that is an intriguing idea. However, as mentioned, HDI and GDP are correlative rather than causative indicators which represent slowly evolving factors such as infrastructure, legislation, and public awareness. As such, in reality we wouldn't expect an immediate short-term response to say, a financial crash. On the other hand, in the model we would see such a response if, for example, GDP dropped sharply and significantly. This would probably be unrealistic, although it is possible that immediate cuts to public services may resulting less effective fire fighting that year. For this reason, HDI (which is less sensitive to short term financial events) may be a more robust metric.

We propose to remove the existing sentences from line 604 to 607 and replace them with the following:

"There are other issues that might arise with using GDP/HDI. We note that there is greater variation of HDI and GDP in space than in time, and it is likely that the spatial variation dominates the fitted model response. However, introducing HDI does allow the model to capture the declining rend in NCV burning (Fig. F3) so the temporal response seems to be reasonable. Another potential issue is that annual GDP is sensitive to short term financial crises or other abrupt changes. Such a drop in GDP would have an immediate effect in the model and this is likely not entirely realistic (although we are unaware of any studies attempting to quantify this). HDI is likely a better indicator in this regard as economic activity comprises only one third of its value, the other two factors (life expectancy and years in education) are not so immediately susceptible to short term changes in economic circumstances."

- L655: Did the authors consider other remote-sensed burned products that might include small fires like GFED4s?

Actually, ESA FireCCI51 also includes enhanced sensitivity to small fires and results in a similar burnt area to GFED4s (~450 Mkm$^2$, Lizundia-Loiola et al. 2020). We considered using GFED5, but by the time it was released our study was fairly advanced and we also became aware of an issue whereby the product overpredicts significantly in Sweden, which would be very problematic for our study domain. As we already discuss the small fires issues in broad terms, so we propose to simply mention the enhanced FireCCI51 small fire sensitivity by modifying the sentence at line 654 to read:

"Remote sensing products based on MODIS (including the ESA FireCCI51 data used here) are known to struggle with detecting small fires and have high omissions errors, particularly in the

Mediterranean (Katagis and Gitas, 2021), although FireCCI51 does feature improved sensitivity to small fires (Lizundia-Loiola et al., 2020). "

- L690: Finally, can the authors address if this model is specific to this region or could be transferred to other regions of the world? How involved do they believe the process of doing this would be?

We believe the actual model is very specific to Europe but the methodology and many outcomes can be taken to other regions. We have some preliminary work for NCV fires globally which shows promise. Cropland fires will likely be trickier and may need to take into account cropland specific management practices, and the grassland fires may also need to be handled explicitly (but that is also not confirmed at this stage). Doing regional studies would likely be far more tractable than global, especially for agricultural fires. We propose to add a new closing sentence to the Conclusions:

"In addition, the scientific outcomes and methodology developed here can facilitate the development of similar models for other regions."

Minor comments:

- L26 here and elsewhere rephrase meteorological fire danger for clarity

We actually only use that phrase in the abstract and suggest changing it to: "fire weather danger".

- L34 remove "of" just before "state of the art"

Yes, but actually we changed it to "to", because we do need the appropriate preposition there.

- L39-40 suggest rephrasing these sentences

We suggest to change it to:

"The strong model skill of BASE when reproducing seasonal and interannual dynamics of NCV burning and the novel inclusion of cropland burning indicate that BASE is well suited for integration in land surface models."

- L49-52 split and shorten this sentence

Yes, it is a bit long. We suggest:

"It interacts with many components of the Earth system, with notable effects on biogeochemical cycles, surface energy budgets, and vegetation dynamics and composition (Archibald et al., 2018; Bowman et al., 2009). Through these effects, fire alters the chemical composition of the atmosphere and the physical properties of the land surface, thereby influencing regional and global climate (Archibald et al., 2018; Bowman et al., 2009; Jones et al., 2022)."

- L58 rephrase "coherent political level" for clarity

We have replaced "coherent political level over a broad spatial extent." With:

"a local, national and transnational levels."

- L210 revise "artefacts"

Replaced with "anomalous values"

- Figure 1: Here and through the figures would be clearer if the acronyms were defined in the axis labels and figure captions

Yes, we added explanations of the acronyms to the captions, but believe that also adding them also to the axis labels isn't practical given the length of the full names.

- Figure 5: Here and elsewhere the single shared legend and brief caption could be clearer if they provided information about the mean lines, uncertainty regions, etc.

Ah right, the trend lines, sorry for the omission. We have modified the caption labels and added appropriate variants of:

"The trend (calculated with linear regression) is plotted as a straight line with the 95% confidence interval shown as coloured shading."

**References:**

Lizundia-Loiola, J., Otón, G., Ramo, R., & Chuvieco, E. (2020). A spatio-temporal active-fire clustering approach for global burned area mapping at 250 m from MODIS data. *Remote Sensing of Environment*, *236*, 111493.

---

## Author Comment (AC3)

Response to RC2

We thank the reviewer for taking the time to review our manuscript and for their positive comments.

In the following point-by-point response we have kept the reviewer's comment verbatim in black text, our responses are in blue and proposed new text in green. Line numbers refer to the reviewed version of the manuscript. Text insertions into existing text are additionally denoted with underlining.

The authors present an important contribution to a pertinent issue in wildfire modelling, adding an important perspective to the hot topic of the differences between natural vegetation versus cropland wildfire. The model is well-described and rigorous, finding concrete differences between the drivers of burning in cropland and non-cropland vegetation. Performance statistics are largely better than fireMIP models, but similar-to-worse than the other GLM based studies cited in the introduction. However, as this is the only study focussed on Europe and is at a finer resolution than most of the other studies, the only conclusion that can be reached through this comparison is that BASE's performance meets an acceptable threshold for publication.

Whilst the paper's conclusion emphasises predictive applicability to the projection of future burned area, the overall performance is not greater than existing global and regional modelling methods. The crucial contribution of this paper is the use of a twin-approach to cropland and "natural" vegetation wildfires. The authors demonstrate that this can substantially reduce confounding effects in specific drivers (e.g. population density or fire weather) that differ between land-cover types, and that trends in NCV/cropland fires can be disaggregated. The paper thus reflects an important contribution to the next generation of fire models, towards improving overall model performance and also to understanding the complex dual effects of climate warming and land-use change over time.

Yes, we absolutely agree that whilst we judge the performance of BASE to be sufficient for application, the results about cropland vs NCV fires are the bigger contribution.

Specific Comments:

   L95: provide evidence/citation that existing global fire models do in fact perform worse in other biomes due to this training bias. Is this the case for all widely-adopted fire-enable DGVMs?

Good point. We are not aware of any paper that quantifies this (though it is a good idea to write one). Qualitatively, Figure 4 and Table 2. of Hanston et al. 2020 illustrates the nature of the regional discrepancies well. Excluding MC2 and GlobFIRM (which are poor everywhere), all models do a reasonable job in the dry tropics, especially Africa. But model performance is poor and very variable in other regions. All models largely overpredict in extra tropics, most notably in the US (except for ORCHIDEE-SPITFIRE but that models predict very little fire outside of the dry tropics) and the Mediterranean, Middle East and the very arid zones in central Eurasia. Some features of the Eurasian steppe are reproduced by some models, but patterns in the Boreal zone are not well reproduced.

We therefore to propose mention the over prediction in the extra tropics and to support our comment by citing the figure and the table from Hantson et al 2020 in the sentence at line 90:

"However, they have notable regional discrepancies and in particular over-predict burnt area in the extra tropics (see Fig 4 and Table 2 in Hantson et al., 2020), likely because their global focus leaves them unable to resolve regionally-specific processes or phenomena."

L180/L190: explain why MEPI and PHI are defined as relative to GPP maximum month of prior year; as maxima are less stable against interannual variability than, for example, the study-period mean, what was the advantage to this formulation?

Using the maximum to normalise MEPI and PHI is fundamental to the way the indices work. It implicitly accounts for both different seasonal cycles (specifically the length of the growing season) and overall productivity in grid cells. To account for overall productivity, one could use the mean, but this will give numbers higher than 1 which are not so useful in this context and depends on growing season length. The maximum has the advantage that the values will always be between 0 and 1 regardless of growing season length or overall productivity.

To make this a bit more concrete, consider MEPI for an area with a short growing season. If you normalise by the mean, a month with high (near maximum) productivity might have a value of 3. Now consider a month of high productivity in a grid cell with a long growing season, the value might only reach 1.5. But in terms of what them to capture, this difference is not important and actually counterproductive. They are both high productivity months – i.e. not very flammable. So, one grid cell having value of 3 and the other 1.5 introduces a spurious difference between the grid cells and the model will try to fit to that. In fact, we want the same value for all grid cells when they have high (relative) productivity – and that is what using the maximum allows. So, it is essential to use the maximum.

Regarding interannual variability, we did test using a longer time period (25 months instead of 13) to derive the maximum but this didn't improve our results.

So, to give a short explanation to clarify this, we propose to add the following at line 185.

"Using the 13-month maximum accounts for the overall productivity of a grid cell in a manner which is insensitive to the length of the growing season (unlike the annual mean)."

L200/Table 1: In table 1 it is stated that FAPAR12 models fine fuel build up over twelve months. At lower values of fAPAR there is a roughly linear relationship with LAI (LAI ~ - 0.5 ln(1 - fAPAR)). This is only one component of leaf litter accumulation. But this neglects leaf mass per area and leaf lifespan, which can vary substantially between needleleaf/broadleaf/deciduous/evergreen. So what does fAPAR12 actually mean, and why is it selected over the more physical GPP? Could it be that the modelled GPP product does not give as spatially reliable a map as the remotely-sensed fAPAR? This could explain the decreased spatial (but improved interannual) performance when GPP replaces fAPAR (table 2). This issue can either be addressed with a good explanation of FAPAR12's physical effect vs GPP12 in the method section, or by discussing the implications further in section 4.1.3.

This is a good point. The logic behind FAPAR12 was that it should provide a better handle on fine fuel (i.e. leaves) than GPP (some of which will be put into wood), even with the difference in leaf longevity and LMA. But we also tried GPP12 as an alternative, because as you state, it is a little more physical. But then we didn't give too much thought as to why FAPAR12 was the better spatial indicator and gave higher deviance explained even though GPP12 gave better temporal IAV.

Thinking about it now, we think the better spatial performance is probably is indeed because FAPAR12 gives a more direct measure of leaf production (imperfect as it may be). We do have good confidence in the GOSIF GPP product because, being based on SIF and not greenness (like, say MODIS GPP) is much closer to a "direct observation" of GPP. So, we don't wish to attribute the relatively poorer performance of GPP to poorer quality data. We propose to clarify FAPAR12's particular sensitivity to leaf/fine fuel at line 199:

"The fraction of absorbed photosynthetically active radiation (FAPAR) is a proxy for live leaf biomass and can be used to quantify fine fuel buildup and availability (Forkel et al., 2017; Knorr et al., 2016; Kuhn-Régnier et al., 2021)."

And insert at line 475 in the Discussions:

"The better performance of FAPAR12 than GPP12 can be explained by FAPAR's specific relationship to leaf biomass rather than GPP's relationship to general biomass production, and the importance of fine fuel (i.e. leaves) for enabling fire ignition and spread."

Unfortunately, by their construction, the FAPAR12 and GPP12 quantities have both spatial and interannual effects. This makes it very difficult to suggest why GPP12 is better temporally better and prefer not to speculate in this manner. However, it does highlight that there is definitely research to be done here, and that future studies may disentangle this by more carefully considering "spatial only" and "temporal only" predictors.

L214/223: Two sources of GDP data are cited, which was used in this study?

Apologies, this was simply an error in the manuscript. We only used the Kummu data for GDP. We have corrected this.

L478: Could it be that these regions are too wet to seasonally burn?

Yes, indeed that is a reasonable explanation. Ideally the model would account for this directly through the monthly FWI, but this might not fully account for the effect. We propose to add (at line 278):

"or do not have an appropriate burn window due to insufficient precipitation seasonality."

L568: Spain does differ, but there also appears to be more simulated cropland fire in France, Poland and the Baltics. It would be good to either acknowledge this difference, or to justify that this visual difference in the maps is not as significant as the Spanish case (e.g. due to GLM 'smearing' or colormap choice).

Yes, there are overestimates of cropland burning in France, Poland and the Baltics, but these are less than in Spain, and, as the reviewer points out, these are a consequence of the smear of low values from the GLM which are somewhat overemphasised by the threshold in the colour scale. Also, the NCV change is particularly remarkable, we should empathise this more. We propose to clarify by modifying the text at line 559 to read:

"This is particularly clear in the observations when comparing as the observed NCV fires in Spain to neighbouring Portugal (Figure 4). However, BASE NCV fails to simulate this change in fire occurrence at the national border. Furthermore, BASE Cropland also overestimates in Spain, predicting extensive area of cropland burning when in fact there are only limited areas. This overestimate is larger than the low levels of overprediction seen in, for example France and Poland, which is a consequence of the GLM tendency to predict a lot of low values (which is here overemphasized by the threshold in

the colour scale).   These substantial overestimates of both cropland and NCV fire occurrence may indicate phenomena specific to Spain which are not accounted for in BASE."

 L687: Consider changing "so is suitable for projecting changes in fire hazard over annual-to-decadal time scales" to "so is suitable for projecting differing changes in fire hazard between cropland and non-cropland vegetation over annual-to-decadal time scales". Not to say it cannot be used in general projections, but that this is the unique value of the model compared to similarly performing fire models.

Yes, although because we do think the model is generally suitable for projections (especially given its regional focus and higher spatial resolution compared to the global models), we suggest the formulation:

"... so is suitable for projecting changes in fire hazard over annual-to-decadal time scales, particularly when considering cropland and non-cropland land cover types."

Technical Corrections:

   L245-250: give the list of actual land cover classes in the supplementary for better readability.

Yes, good idea, we are happy to do this.

   L393: add full-stop.

Done.

   L490/491: correct "1)" and "2)" to (1) and (2) in sentence.

Done.